# Evaluation of antimicrobial and antiproliferative activities of Actinobacteria isolated from the saline lagoons of northwestern Peru

Rene Flores Clavo[1,2,3]*, Nataly Ruiz Quiñones[2,3], Álvaro Jose Hernández-Tasco[4], Marcos José Salvador[4], Ana Lúcia Tasca Gois Ruiz[5], Lúcia Elaine de Oliveira Braga[6], Jonas Henrique Costa[7], Taícia Pacheco Fill[7], Zhandra Lizeth Arce Gil[8], Luis Miguel Serquen Lopez[3,9], Fabiana Fantinatti Garboggini[2]

1 Graduate Program in Genetics and Molecular Biology, Institute of Biology, University of Campinas (UNICAMP), Campinas, São Paulo, Brazil, 2 Chemical, Biological and Agricultural Pluridisciplinary Research Center (CPQBA), University of Campinas (UNICAMP), Campinas, Paulínia, São Paulo, Brazil, 3 Department of Biotechnology, Research Center and Innovation and Sciences Actives Multidisciplinary (CIICAM), Chiclayo, Lambayeque, Perú, 4 Department of Plant Biology Bioactive Products, Institute of Biology Campinas, University of Campinas, Campinas, São Paulo, Brazil, 5 Faculty Pharmaceutical Sciences, University of Campinas, Campinas, São Paulo, Brazil, 6 Graduate Program in Odontology School of Odontology of Piracicaba, University of Campinas, Campinas, São Paulo, Brazil, 7 Institute of Chemistry, University of Campinas, Campinas, São Paulo, Brazil, 8 Catholic University Santo Toribio of Mogrovejo, Facultity of Human Medicine, Chiclayo, Lambayeque, Perú, 9 Direction of Investigation Hospital Regional Lambayeque, Chiclayo, Lambayeque, Perú

* renefloresclavo@gmail.com

## Abstract

Extreme environments Morrope and Bayovar Salt lagoons, several ecosystems and microhabitats remain unexplored, and little is known about the diversity of *Actinobacteria*. We suggest that the endemic bacteria present in this extreme environment is a source of active molecules with anticancer, antimicrobial, and antiparasitic properties. Using phenotypic and genotypic characterization techniques, including 16S rRNA sequencing, we identified these bacteria as members of the genera *Streptomyces*, *Pseudonocardia*, *Staphylococcus*, *Bacillus*, and *Pseudomonas*. *Actinobacteria* strains were found predominantly. Phylogenetic analysis revealed 13 *Actinobacteria* clusters of *Streptomyces*, the main genus. Three Streptomycetes, strains MW562814, MW562805, and MW562807 showed antiproliferative activities against three tumor cell lines: U251 glioma, MCF7 breast, and NCI-H460 lung (non-small cell type); and antibacterial activity against *Staphylococcus aureus* ATCC 6538, *Escherichia coli* ATCC 10536, and the multidrug resistant *Acinetobacter baumannii* AC-972. The antiproliferative activities (measured as total growth inhibition [TGI]) of *Streptomyces* sp. MW562807 were 0.57 µg/mL, for 0.61 µg/mL, and 0.80 µg/mL for glioma, lung non-small cell type, and breast cancer cell lines, respectively; the methanolic fraction of the crude extract showed a better antiproliferative activity and could inhibit the growth of (U251 (TGI = 38.3 µg/mL), OVCAR-03 (TGI = 62.1 µg/mL), and K562 (TGI = 81.5 µg/mL)) of nine tumor cells types and one nontumor cell type. Extreme enviroments, such as the Morrope and

**Data Availability Statement:** The data for the 13 isolates that were identified in the Morrope and Bayovar salt saloons can be accessed at the following links: 1. https://www.ncbi.nlm.nih.gov/nuccore/MW562805.2 2. https://www.ncbi.nlm.nih.gov/nuccore/MW562806.2 3. https://www.ncbi.nlm.nih.gov/nuccore/MW562807.2 4. https://www.ncbi.nlm.nih.gov/nuccore/MW562808.2 5. https://www.ncbi.nlm.nih.gov/nuccore/MW562809.2 6. https://www.ncbi.nlm.nih.gov/nuccore/MW562812.2 7. https://www.ncbi.nlm.nih.gov/nuccore/MW562813.2 8. https://www.ncbi.nlm.nih.gov/nuccore/MW562814.2 9. https://www.ncbi.nlm.nih.gov/nuccore/MW562815.2 10. https://www.ncbi.nlm.nih.gov/nuccore/MW562816.2 11. https://www.ncbi.nlm.nih.gov/nuccore/MW562817.2 12. https://www.ncbi.nlm.nih.gov/nuccore/MW562810.2 13. https://www.ncbi.nlm.nih.gov/nuccore/MW562811.2 Data underlying this study can also be found at https://gnps.ucsd.edu/ProteoSAFe/result.jsp?task=7bdaa9dfa79e46b695c662fc073cc6e0&view=group_by_compound.

**Funding:** This study has been financed by the Concytec - World Bank Project "Improvement and Expansion of the Services of the National System of Science, Technology and Technological Innovation" 8682-PE, to the "World Bank", to "CONCYTEC" through its executing unit ProCiencia [contract number 190-2018]" and "The postgraduate programs in Genetics and Molecular Biology.

**Competing interests:** NO authors have competing interests The authors have declared that no competing interests exist.

Bayovar Salt saloons are promising sources of new bacteria, whose compounds may be useful for treating various infectious diseases or even some types of cancer.

## Introduction

Several regions in Peru have extreme environments, such as the salt marshes located on the coast, center, and south of the country. These regions, have several unexplored ecosystems and microhabitats, and consequently limited reports on the diversity of bacteria and other organisms [1–3]. Among microorganisms, members of the phylum *Actinobacteria* can be found in all types of extreme environments. *Micromonospora*, *Actinomadura*, and *Nocardiopsis* have been reported from saline soils of ephemeral salty lakes in Buryatiya [4], whereas *Streptomyces*, *Nocardiopsis*, and *Nocardioides* have been isolated from the Western Ghats region of India [5]. *Micromonospora*, *Streptomyces*, *Salinispora*, and *Dietzia* have been isolated from the coastal zone in Chile [6]. Additionally, halophilic and halotolerant strains of Actinobacteria show heterogeneous physiological characteristics for different genera because these bacteria can synthesize secondary metabolites to cope with the high salinity and extreme temperature conditions of their environments [7–9]. These extreme conditions favor the development of metabolic competitiveness to the production enzyme, which can bacterial population adapt to high salinity [10]. Furthermore, these microorganisms can perform essential processes, such as carbon cycle, metal transfer, and the removal of organic pollutants at higher trophic levels [11, 12]. Interestingly, this microbial group presents a unique ability to produce new products, mainly antibiotics [13, 14]. However, information on substances isolated from microorganisms inhabiting saline environments is scarce.

The search for these microorganisms has been mainly associated with the production of antibiotics and antitumor substances [15]. Approximately 22,500 biologically active substances are obtained from microorganisms, 45% of which are represented by Actinomycetes, and 70% are *Streptomyces*-derived [16, 17]. According to Lam (2006), new Actinobacteria from these unexplored habitats could serve as the source of new bioactive secondary metabolites [18].

The bio-guided study strategy helped us identify several different microorganisms with biotechnological potential [19]. In this study, we performed a phylogenetic analysis of a collection of bacterial isolates from the saline lagoons of northwestern Peru. Furthermore, we explored their potential as producers of secondary metabolites with antimicrobial and antiproliferative activities. we have revealed that the Actinobacteria existing in these extreme environmental conditions have diverse characteristics and could comprise new species that produce novel and biologically active compounds.

## Materials and methods

### Site description and sampling

Samples were collected from zone "1", zone "2", and zone "3" of Morrope saline lagoons in December 2012, January 2013, December 2014, respectively; and zone "4") at Bayovar saline lagoons in March 2015 (Fig 1). This study did not require permission from a competent authority since it excludes any protected species; besides, documentation is currently being processed to achieve its registration. The sampling areas were the lagoons, and sampling was performed based on the Nagoya Protocol.

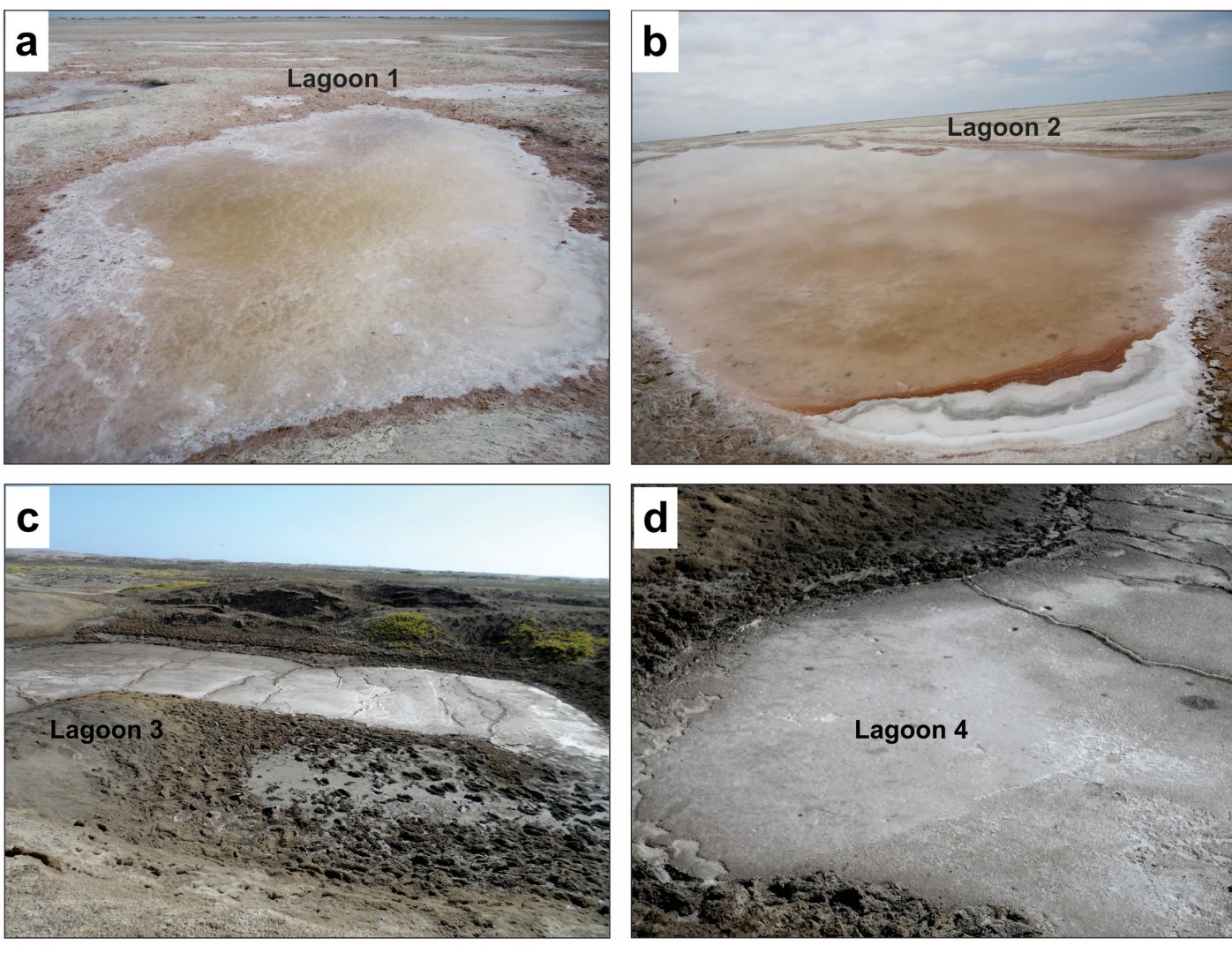

**Fig 1.** Sampling sites: a, b, c = lagoons 1, 2, and 3 (State Morrope); and d = lagoon 4 (State Bayovar).

Table 1 indicates the sources and locations from which samples were obtained. Samples were collected aseptically, placed in sterile plastic bags, and refrigerated at 4°C for further processing in the Bacteriology Laboratory, Regional Hospital of Lambayeque. A flowchart depicting the methodological strategy adopted in this study is shown (S1 Fig).

**Table 1. Data related to saline lagoon samples.**

| Morrope lagoons | | | | |
|---|---|---|---|---|
| Collection zone | Sample type | Geographic location | | |
| Zone 1 (December 2012) | A; AES | 6°10S;80°35W (3 samples × triplicate) | 6°11S;80°37W (3 samples × triplicate) | 6°9S;80°39W (3 samples × triplicate) |
| Zone 2 (January 2013) | AES; SE | 6°19S;80°28W (3 samples × triplicate) | 6°17S;80°26W (3 samples × triplicate) | 6°18S;80°27W (3 samples × triplicate) |
| Zone 3 (December 2014) | A; SE | 6°08S;80°50W (3 samples × triplicate) | 6°08S;80°51W (3 samples × triplicate) | 6°08S;80°40W (3 samples × triplicate) |
| Bayovar lagoons | | | | |
| Zone 4 (March 2015) | A; AES; SE | 6°23S;80°66W (3 samples × triplicate) | 6°25S;80°26W (3 samples × triplicate) | 6°23S;80°78W (3 samples × triplicate) |

A = saline water, AES = water and sediment, SE = sediment.

## Isolation of cultivable Actinobacteria

A 10.0 mL aliquot of saline lagoon sample was transferred to an Erlenmeyer flask containing 10.0 mL of salt broth (0.6% (w/v) yeast extract Difco, 2% (w/v) glucose Merck, 5% (w/v) peptone Merck, 3% (w/v) meat extract Difco supplemented with chloramphenicol Medrock, and 1% (w/v) fluconazole Genfar at pH 7.0). Erlenmeyer flasks were homogenized and maintained at 50˚C in a water bath for 60 min to reduce the pollutant load [20, 21], as described in a previously reported protocol [22]. The isolates were reactivated and stored in a refrigerated chamber at -20˚C for transporting to the Microbial Resources Division Laboratory of the Pluridisciplinary Center for Chemical, Biological and Agricultural Research (CPQBA) in August 2015.

## Identification of bacterial isolates

**Reactivation and morphological phenotypic characterization of bacterial strains.** The reactivation of the isolates bacterial was performed as previously described [22], the isolates from the Culture Collection of Research Center and Innovation and Sciences Actives Multidisciplinary (CIICAM), Department of Biotechnology, Chiclayo, Lambayeque, Perú; these were labeled with location codes according to the salt saloons from which they were collected, for example (B-81), which refers to the Bayovar saline saloon sample 8 culture 1.

**DNA extraction and 16S rRNA gene sequencing.** An isolated colony was used for genomic DNA, according to the method described by Pospiech and Neuman [23], with some modifications according to a previously published protocol [24]. The Amplification of the 16S rRNA gene was performed via polymerase chain reaction (PCR). The specifications of this protocol have been previously described [25]. The samples were purified using minicolumns GFX PCR DNA & gel band purification kit (GE Healthcare Bio-Sciences AB Uppsala, Sweden), according to a previously described protocol [26], and sequenced using an ABI3500XL Series automatic sequencer (Applied Biosystems Foster City, California, USA) in the Laboratory of the Division of Microbial Resources, Chemical, Biological and Agricultural Pluridisciplinary Research Center (CPQBA), University of Campinas (UNICAMP), Paulínia, Sao Paulo, Brazil). Sequencing reactions were performed using the Big Dye Terminator Cycle Sequencing Ready Reaction Kit (Applied Biosystems) according to the manufacturer's modified by the authors protocol [27]. Partial sequences of the 16S ribosomal RNA gene obtained from each isolate were assembled into a contig using BioEdit 7.0 [28]. The sequences of organisms were added to the EZBioCloud 16S Database (https://www.ezbiocloud.net/)) using the "Identify" service [29], and species assignment was based on the closest hits [30]. 16S rRNA gene sequences retrieved from the database and related to the unknown organism gene were selected for alignment in the Clustal X program [31], and phylogenetic analyses were performed using the MEGA version 7.0 program [32]. The evolutionary distance matrix was calculated using the Kimura-2 model parameters [33], and the phylogenetic tree was constructed from the evolutionary distances calculated by the neighbor-joining method [34], with bootstrap values based on 1000 resamples.

## Therapeutic potential of bacterial isolates

**Crude extract produced from extremophilic bacteria.** The evaluation of the secondary metabolites of bacterial isolates extracted with ethyl acetate was performed in three culture media—R2A broth (Himedia ref. 1687), ISP2 broth (Difco ref 277010), and nutrient broth (Termofisher Scientific ref CM0001)—as described in a previously published protocol [35]. The crude extracts of three representative strains were used for antimicrobial and antiproliferative activity tests. Simultaneously, the crude extracts were analyzed by ultra-high pressure

liquid chromatography-mass spectrometry (UHPLC-MS) in a Thermo Scientific QExactive® Hybrid Quadrupole-Orbitrap Mass Spectrometer.

**Fractionation of the crude extract from extremophilic bacteria.** The crude extracts from the isolates were fractionated using a previously described protocol [35], and the fractions were tested for their antibacterial and antiproliferative activities.

***In vitro* antibacterial activity assay.** Three crude extracts from *Streptomyces* sp. MW562814, MW562807, and MW562805 were tested for antimicrobial activity using the minimum inhibitory concentration (MIC) assay, following the protocol reported by Siddharth and Vittal [36]. The crude extracts were partially diluted in 1% dimethyl sulfoxide (DMSO; 0,39mg/mL–1 mg/mL), and the microbial cultures were grown in sterile broth to obtain a total volume of 200 μL. The 96- well plate was incubated at 37˚C (room temperature); the lowest concentration of the extract, that completely inhibited bacterial growth was considered the MIC. Each biological assay was performed in triplicate. The pathogenic bacteria used in this test were *Escherichia coli* ATCC 10536, *Staphylococcus aureus* ATCC 6538, and *Acinetobacter baumannii* AC-972. The source strain was obtained from the bank of multidrug-resistant (MDR) isolates from a patient with pneumonia from the Intensive Care Unit (ICU) of the Hospital Regional Lambayeque. Patient data linked to the samples were anonymized for access.

***In vitro* antiproliferative activity assay.** This assay aimed detected anticancer activities by evaluating antiproliferative activity against human tumor cells [37]. *In vitro* tests assessing the effects of the crude extract of *Streptomyces* sp. MW562805, MW562807, and MW562814 on human tumor cell lines of different origins and a non-tumor cell line were performed. Human tumor cell lines [U251 (glioblastoma), UACC-62 (melanoma), MCF-7 (breast, adenocarcinoma), NCI-ADR/RES (multi-drug resistant ovarian adenocarcinoma), 786–0 (renal, adenocarcinoma), NCI-H460 (lung, non-small cell carcinoma), PC-3 (prostate, adenocarcinoma), OVCAR-03 (ovarian, adenocarcinoma), and K562 (chronic myeloid leukemia)] were kindly donated by Frederick Cancer Research & Development Center, National Cancer Institute, Frederick, MA, USA. One immortalized cell line (HaCaT, human keratinocyte) was kindly donated by Dr. Ricardo Della Coletta (University of Campinas). Stock cultures were grown in a medium containing 5 mL of RPMI 1640 (GIBCO BRL, Gaithers-Burg, MD, USA) supplemented with 5% (v/v) fetal bovine serum (GIBCO) at 37˚C and 5% (v/v) $CO_2$. Penicillin:streptomycin ($1000 \ \mu g \cdot L^{-1}$:$1000 \ U \cdot L^{-1}$, $1 \ mL \cdot L^{-1}$, Vitrocell, Campinas, SP, Brazil) was added to the experimental cultures. Cells in 96-well plates ($100 \ \mu L$ cells $well^{-1}$) were exposed to the extracts in DMSO (Sigma-Aldrich)/RPMI (0.25, 2.5, 25, and 250 μg·mL−1) at 37˚C and 5% (v/v) $CO_2$ for 48 h. The DMSO final concentration of 0.2% (w/v) did not affect cell viability. Before ($T_0$) and after ($T_1$) sample application, cells were fixed with 50% (w/v) trichloroacetic acid (Merck), and cell proliferation was determined by the spectrophotometric quantification (540 nm) of cellular protein content using the sulforhodamine B assay. Verified in the concentration-response curve for each cell line, the values of the sample concentration required to produce total growth inhibition (TGI) or cytostatic effect through non-linear regression analysis using ORIGIN 8.6® (OriginLab Corporation, Northampton, MA, USA).

## Mass spectrometry analysis

*Streptomyces* sp. MW562807 extract was resuspended in 1 mL of methanol (HPLC grade), and 100 μL of this suspension was diluted in 900 μL of methanol to a final concentration of 6.4 mg/mL. (UHPLC-MS) analyses were performed using a Thermo Scientific QExactive® Hybrid Quadrupole-Orbitrap Mass Spectrometer according to a previously published protocol [38].

## Molecular MS/MS network

A molecular network for *Streptomyces* sp. MW562807 was constructed using the online workflow at Global Natural Products Social Molecular Networking (GNPS) (https://gnps.ucsd. edu/). The library spectra were filtered in the same manner as the input data. All matches maintained between network and library spectra were required to have a score above 0.5 and at least five matched peaks [39]. These procedures were performed using a previously described protocol [38].

## Statistical analyses

The data are expressed as the mean ± standard error of the mean (SEM). Statistical comparisons were performed using a one-way ANOVA followed by the Student–Newman–Keuls test, and the differences were considered statistically significant when $P < 0.05$ and were determined through non-linear regression analysis using ORIGIN 8.6® (OriginLab Corporation, Northampton, MA, USA).

## Results and discussion

### Isolation, identification, and selection of bacterial species from the northern saline lagoons of Peru

In total, 50 pure cultures showing different colony morphologies were obtained, which were grown on R2A medium containing saline water (A) only, water and sediment (AES), and sediment (SE) only (Table 2). Isolates were clustered into 42 filamentous and 8 non-filamentous bacterial groups with similar characteristics based on morphological characteristics, including the aerial mycelium, morphological spore mass color, pigmentation of vegetative or substrate mycelium, and the production of diffusible pigment [40].

The isolates were regrouped into their similar phenotypic characteristics and identified based on the sequencing and alignment from the 16S rRNA gene analyses. In total, 13 bacterial isolates from the saline lagoons of northwestern Peru (Morrope and Bayovar) cultivars were identified after the 16S rRNA gene sequencing (Fig 2).

Therefore, these strains should be subjected to further taxonomic and analytical chemistry analyses to confirm their novelty at the species level and as a source of novel chemical entities.

According to EZBioCloud, 13 isolates were identified in the Morrope and Bayovar salt saloons, of wich eight and five groups of *Streptomyces* were identified in the Morrope salt flats only and the Bayovar salt flats, respectively. Three isolates were similar to *S. olivaceus* NRRL B-3009: MW562807 (99.93%), MW562805 (99.84%) (Fig 2), and MW562808 (97.98%) MW562806 was similar to *S. hyderabadensis* OU-40 (98.06). Addionally, MW562809 presented higher similarity with *S. pactum* NBRC 13433 (97.98%); MW562810 and MW562811 presented high similarity to *S. griseorubens* NBRC 12780 and *S. labedae* NBRC 15864 with 97.57% and 97.58% similarity, respectively. MW562812 and MW562813 were similar to *S.*

**Table 2. 16S rRNA gene-based identification of bacteria isolated from lagoons Morrope and Bayovar.**

| Sample type | Number of isolates (%) | Phylum | Family | Genus |
|---|---|---|---|---|
| AES, SE | 39 (78%) | Actinobacteria | *Streptomycetaceae* | *Streptomyces* |
| SE | 3 (6%) | Actinobacteria | *Pseudonocardiaceae* | *Pseudonocardia* |
| A | 2 (4%) | Firmicutes | *Bacillaceae* | *Bacillus* |
| A | 4 (8%) | Firmicutes | *Staphylococcaceae* | *Staphylococcus* |
| AES | 2 (4%) | Proteobacteria | *Pseudomonadaceae* | *Pseudomonas* |

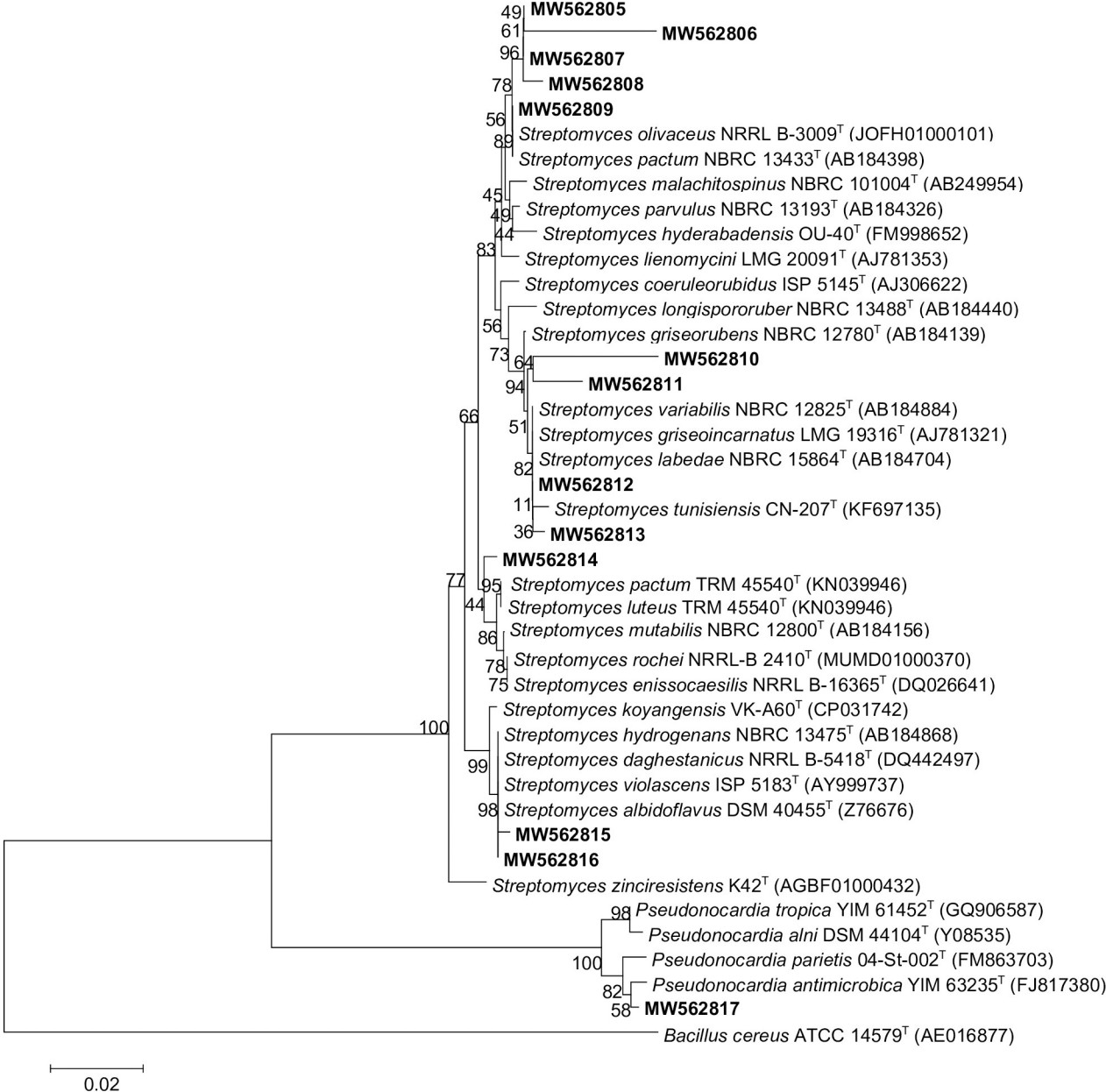

**Fig 2. Phylogenetic tree based on 16S rRNA gene sequences (1206–1500 bp positions in the final dataset) inferred using the neighbor-joining method in MEGA7, the evolutionary distances were computed using the Kimura 2-parameter method, showing the phylogenetic positions of different saline isolates and type strains within the *Streptomyces* and *Pseudonocardia* genera.** Numbers at branching points refer to percentages of bootstrap values from 1000 replicates. Bar, 0.02 substitutions per nucleotide position was used. *Bacillus cereus* ATCC 14579[T] (AE016877) was the outgroup.

*griseoincarnatus* LMG 19316 (97.58%), and *S. variabilis* NBRC 12825 (97.58%), respectively; the latter is another actinobacterium recovered from the Morrope salt saloons. The isolates MW562814, MW562815, MW562816, and MW562817 exhibited similarity with *S. luteus* TRM 45540 (99.16%) (Fig 2), *S. violascens* ISP 5183 (97.37%), *S. hydrogenans* NBRC 13475 (97.27%), and *Pseudonocardia antimicrobica* YIM 63235 (98.41%), respectively. This study is the first to report this genus of *Streptomyces* present in the Bayovar and Morrope saline

**Table 3. Minimum inhibitory concentration of three *Streptomyces* sp. crude extracts, as determined by broth dilution method.**

| Isolates of *Streptomyces* sp. | Minimum inhibitory concentration (µg/mL) | | | |
|---|---|---|---|---|
| | *E. coli* ATCC 10536 | *S. aureus* ATCC 6538 | *A. baumannii* AC-972 MDR | Antibiotic Chloramphenicol |
| MW562814 | 7.82 | 7.82 | 15.63 | 22.2 |
| MW562805 | 15.63 | 15.63 | 15.63 | 23.1 |
| MW562807 | 7.82 | 7.82 | 7.82 | 22.3 |

lagoons. Due to the physicochemical characteristics of the environment, they are considered extremophilic bacteria; however, a different genus such as *Pseudonocardia*, has been reported by Zhang et al. (2016) as producers of γ-butyrolactone molecules present in *Streptomyces* genera [41]. Our culture-based approach using pre-enrichment to decrease the bacterial load corresponds to the reports described in literature, in which the isolation of halophilic bacteria reveals low species richness and dominance of the genus *Streptomyces* [42]. Ballav et al. (2015) reported *Streptomyces* especies as the most predominant group contributing to 46% of the total isolates in crystallizer pond sediments of Ribandar saltern in Goa, India [43]. In this study, isolation was performed using three types of culture media and four different concentrations of salt in the lagoons because the concentration of salts ranged from 0 to 300 psu. Cortés-Albayay et al. (2019) in Salar de Huasco reported similar results; they isolated *Streptomyces* (86%), *Nocardiopsis*, (9%), *Micromonospora* (3%), *Bacillus* (1%), and *Pseudomonas* (1%) [44].

## Antibacterial activity screening

The MIC assay was used to screen antibacterial metabolite-producing strains against the indicator strains *Escherichia coli* ATCC 10536, *Staphylococcus aureus* ATCC 6538, and *Acinetobacter baumannii* AC-972 MDR. The crude extracts tested from the three Actinobacteria were effective against the pathogenic bacteria tested. The antimicrobial activity of the crude extract of *Streptomyces* sp. MW562807 crude extract exhibited the highest inhibitory activity with the lowest MIC (7.82 µg/mL) against the three pathogens, followed by *Streptomyces* sp. MW562814, which showed an MIC of 7.82 µg/mL against *E. coli* ATCC 10536 and *S. aureus* ATCC 6538 and MIC of 15.63 µg/mL against *A. baumannii AC-972* MDR. MW562805 showed an MIC of 15.63 µg/mL against the pathogens tested. Notably, all crude extracts of the strains presented low MIC compared with the reference antibiotic (Table 3).

Similar results were reported by Siddharth & Vidal (2018) [36] for a marine *Streptomyces* sp. S2A with an MIC of 31.25 µg/mL against *Klebsiella pneumoniae*, 15.62 µg/mL against *Staphylococcus epidermidis*, *Staphylococcus aureus*, *Bacillus cereus*, *Escherichia coli*, and *Micrococcus luteus* with 7.8 µg/mL of the *Streptomyces* sp. MW562807 showed an MIC of 7.82 µg/mL against *Escherichia coli*, *Staphylococcus aureus* and *A. baumannii*. The crude extract of *Streptomyces* sp. YBQ59 showed MICs ranging from 10.5 to 22.5 µg/mL, against nine pathogens, which were similar to those we obtained [45].

## Antiproliferative activity screening

Among all isolated microorganisms, crude extracts could be prepared from 23 isolated microorganisms. These extracts were evaluated against a panel of three human tumor cell lines. Only the crude extracts of *Streptomyces* sp. MW562805, MW562807, and MW562814 showed moderate antiproliferative effects, indicating the concentration required to elicit a TGI (Table 4).

According to previous studies [46, 47], TGI values higher than 50 µg/mL were exhibited by inactive samples. The most active extract was *Streptomyces* sp. MW562807 with TGI ranging

**Table 4. Antiproliferative activity of three *Streptomyces* extracts against human tumor cell lines.**

| Crude extract and positive control | TGI (µg/mL) [a] | | |
|---|---|---|---|
| | U251 [b] | MCF-7 [b] | NCI-H460 [b] |
| Doxorubicin [c] | 2.5 ± 2.2 (P) | 6.11* (P) | <0.025 (P) |
| *Streptomyces* sp. MW562814 | 5.1 ± 2.8 (P) | 8.0 ± 3.6 (M) | 9.3 ± 6.9 (M) |
| *Streptomyces* sp. MW562805 | 5.4 ± 4.0 (P) | 8.0 ± 1.5 (M) | 5.1 ± 4.0 (P) |
| Doxorubicin [c] | 1.5 ± 1.2 (P) | >25 (W) | 1.8 ± 1.1 (P) |
| *Streptomyces* sp. MW562807 | 0.57 ± 0.05 (P) | 0.8 ± 0.2 (P) | 0.6 ± 0.1 (P) |

a) Results expressed as concentration required to elicit total growth inhibition (TGI) in µg ml$^{-1}$ followed by standard error, calculated by sigmoidal regression using Origin 8.0 software; Results classified according to CSIR's criteria: inactive (I, TGI ≥ 50 µg ml$^{-1}$), weak (W, 15 µg ml$^{-1}$ ≤ TGI < 50 µg ml$^{-1}$), moderate (M, 6.25 µg ml$^{-1}$ ≤ TGI < 15 µg ml$^{-1}$) or potent (P, TGI < 6.25 µg ml$^{-1}$) activity [46]

* Estimated TGI value = when experimental data did not converge (standard error higher than calculated effective concentration).

b) Human tumor cell lines: U251 (glioblastoma); MCF7 (breast, adenocarcinoma); NCI-H460 (lung, non-small cell carcinoma).

c) Doxorubicin (positive control, 0.025 to 25 µg ml$^{-1}$).

from 0.57 µg/mL (U251, glioblastoma) to 0.80 µg/mL (MCF-7, mammary adenocarcinoma) (Table 4).

Based on these results, *Streptomyces* sp. MW562807 strain was cultured in three media to evaluate the influence of nutrient conditions on the production of bioactive compounds. The nutrient broth growth medium (NA) yielded an less active extract, while ISP2 and R2A media yielded active extracts with different profiles (Table 5).

The crude extract of *Streptomyces* sp. MW562807 in the ISP2 medium moderately inhibited the proliferation of glioblastoma (U251, TGI = 14.1 µg/mL) and melanoma (UACC-62, TGI = 17.2 µg/mL) cell lines. In contrast, the crude extract of *Streptomyces* sp. MW562807 in the R2A medium potently inhibited glioblastoma cell growth (U251, TGI = 1.6 µg/mL) along with moderate anti-proliferative effects against lung carcinoma (NCI-H460, TGI = 11.2 µg/mL) and leukemia (K562, TGI = 11.6 µg/mL) cell lines. Despite the expectation of a fractionation-mediated increase in antiproliferative activity, fractions 1–6 showed lower antiproliferative activity. The best activity was observed for fraction 3, which weakly inhibited the proliferation of U251 (TGI = 38.3 µg/mL glioma) cells (Table 5). Besides synergism and antagonism, the combined effect of natural products has been extensively studied over the years. Many studies have demonstrated that the crude extract exhibits more (synergistic) or less (antagonistic) effects than less complex fractions or even the isolated compounds [48–50].

Due to its metabolic and genetic capacities, several secondary metabolites produced by *Streptomyces* species have shown anti-proliferative activity [51]. Furthermore, throughout the evolutionary history of aquatic organisms and as a response to the pressure of environmental selection, actinomycetes isolated from marine environments have a greater capacity to express secondary metabolites with unique chemical structures [18]. This pressure has generated high specificity and a complex three-dimensional conformation of the compounds to act in the marine environment [52]. In this particular case, the geographical isolation, geological conditions of the saline lagoon formation, and extreme environmental generate a greater environmental selection pressure, which enables the analysis of antibiotic and antiproliferative activity in the isolates.

The anticancer potential of halophilic actinomycetes has been extensively studied. Secondary metabolites Salternamides A-D isolated from halophilic actinomycetes of saline on the island of Shinui (Republic of Korea) have potential cytotoxicity against human colon cancer (HCT116) and gastric cancer (SNU638) cell lines [53]. Another moderately halophilic

**Table 5. Antiproliferative evaluation of isolate *Streptomyces* sp. MW562807 fermented in three different media and fractions of the R2A medium against nine tumors and one non-tumor cell line.**

| Cell lines [b] | TGI (µg/ml) [a] | | | | | | | | | |
|---|---|---|---|---|---|---|---|---|---|---|
| | Dox [c] | MW562807 [d] | | | MW562807 R2A Fractions [e] | | | | | |
| | | NB | ISP2 | R2A | 1 | 2 | 3 | 4 | 5 | 6 |
| U251 | 0.04 ± 0.03 (P) | 44.7 ± 29.7 (W) | 14.1 ± 5.2 (M) | 1.6 ± 1.5 (P) | # | # | 38.3 ± 23.4 (W) | 73.5 ± 36.6 (I) | # | 163.0 ± 109.6 (I) |
| UACC-62 | 0.8 ± 0.3 (P) | 250 (I) | 17.2 ± 5.1 (M) | 21.3 ± 17.2 (W) | # | # | 127.2 ± 21.5 (I) | 89.4 ± 31.6 (I) | # | # |
| MCF-7 | >25 (W) | # | # | 8.0* (M) | # | # | 154.6 ± 53.4 (I) | 184.5 ± 23.7 (I) | # | # |
| NCI-ADR/RES | >25 (W) | # | 155.9 ± 65.2 (I) | 31.0 ± 10.6 (W) | # | # | 189.8 ± 118.1 (I) | 143.2 ± 43.1 (I) | # | # |
| 786-0 | 0.2 ± 0.1 (P) | # | * | 39.0 ± 35.1 (W) | # | # | 172.6 ± 162.8 (I) | # | # | # |
| NCI-H460 | >25 (W) | # | 38.9 ± 11.0 (W) | 11.2 ± 6.2 (M) | # | # | 108.0 ± 16.0 (I) | 235.1 ± 26.7 (I) | # | # |
| OVCAR-03 | 14.4 ± 6.8 (P) | # | 55.2 ± 18.0 (I) | 19.4 ± 3.5 (W) | # | # | 62.1 ± 30.0 (I) | 81.6 ± 32.1 (I) | # | # |
| HT29 | >25 (W) | # | 155.7 ± 147.4 (I) | 89.7 ± 72.5 (I) | # | # | 167.4 ± 33.5 (I) | 195.2 ± 37.8 (I) | # | # |
| K562 | 6.3 ± 2.2 (P) | # | # | 11.6 ± 7.3 (M) | # | # | 81.5 ± 26.8 (I) | 100.4 ± 78.3 (I) | # | # |
| HaCat | >25 (W) | # | # | >250 (I) | # | # | # | # | # | # |

a) Results expressed as concentration required to elicit total growth inhibition (TGI) in µg ml$^{-1}$ followed by standard error, calculated by sigmoidal regression using Origin 8.0 software; Results classified according to CSIR's criteria: inactive (I, TGI ≥ 50 µg ml$^{-1}$), weak (W, 15 µg ml$^{-1}$ ≤ TGI < 50 µg ml$^{-1}$), moderate (M, 6.25 µg ml$^{-1}$ ≤ TGI < 15 µg ml$^{-1}$) or potent (P, TGI < 6.25 µg ml$^{-1}$) activity [46]

* Estimated TGI value = when experimental data did not converge (standard error higher than calculated effective concentration)

# TGI > 250 µg ml$^{-1}$.

b) Human tumor cell lines: U251 (glioblastoma), UACC-62 (melanoma), MCF-7 (breast, adenocarcinoma), NCI-ADR/RES (multi-drug resistant ovarian adenocarcinoma); 786-0 (kidney, adenocarcinoma); NCI-H460 (lung, non-small cell carcinoma); OVCAR-3 (ovarian adenocarcinoma); HT29 (colon, adenocarcinoma); and K562 (chronic myeloid leukemia). Human non-tumor cell line: HaCaT (immortalized keratinocytes).

Samples

c) Dox = doxorubicin (positive control)

d) MW562807 = *Streptomyces* sp. crude extract in nutrient broth growth medium (NB), yeast extract-malt medium (ISP2), or R2A growth medium (R2A)

e) MW562807 R2A fractions: 1 = H20; 2 = MeOH-H$_2$O; 3 = MeOH; 4 = MeOH–EtOAc; 5 = 100% EtOAc; 6 = EtOAc-acid (MW562807 R2A crude extract submitted to vacuum chromatographic C18-column).

*Streptomyces* sp. Nov WH26 with the GenBank accession number JN187420.1, isolated from Weihai Solar Saltern in China, exhibits potent cytotoxicity against human cancer cell lines of the lung (A549, IC$_{50}$ 78.6 µM), cervical epithelium (HeLa, IC$_{50}$ 56.6 µM), liver cancer cell (BEL-7402, IC$_{50}$ 47.1 µM), and colon (HT-29, IC$_{50}$ 94.3 µM) [54]. Four compounds, shellmicin A-D isolated from the *Streptomyces* sp. shell-016 were described as cytotoxic agents against five human tumor cell lines: non-small cell lung cancer (H1299, ATCC CRL-5803), malignant melanoma (A375, ATCC CRL- 1619), hepatocellular carcinoma (HepG2, ATCC HB-8065), colorectal adenocarcinoma (HT29 ATCC HTB-38), and breast cancer (HCC1937, ATCC CRL-2336) [55]. Moreover, the compounds Shellmicin A, B, and D showed greater cytotoxicity (IC$_{50}$ ranging from 0.69 µM to 3.11 µM at 72 h) than Shellmicin C. Interestingly, shellmicin C and D are a pair of stereoisomers, and their biological activity differ considerably.

Two active substances Chromomycin SA and 1-(1H-indol-3-yl)-propane-1,2,3-triol, were isolated and identified from the actinomycete strain *Streptomyces* sp. KML-2 wich was isolated from a saline soil mine in Khewra, Pakistan, Both substances, could reduce Hela cell viability (human cervical cancer) in a similar magnitude (IC$_{50}$ 8.9 and 7.8 µg/mL, respectively) while MCF-7 cells (human breast cancer) were more sensitive to 1-(1H-indol-3-yl)-propane-1,2,3-triol (IC$_{50}$ 0.97 µg /mL [56]. This study also revealed that the Khewra salt mine from which the KML-2 strain was isolated is a powerful ecological niche with inimitable strain diversity that is yet to be discovered, a fact that reinforces our study approach in the search for new compounds with biotechnological activity.

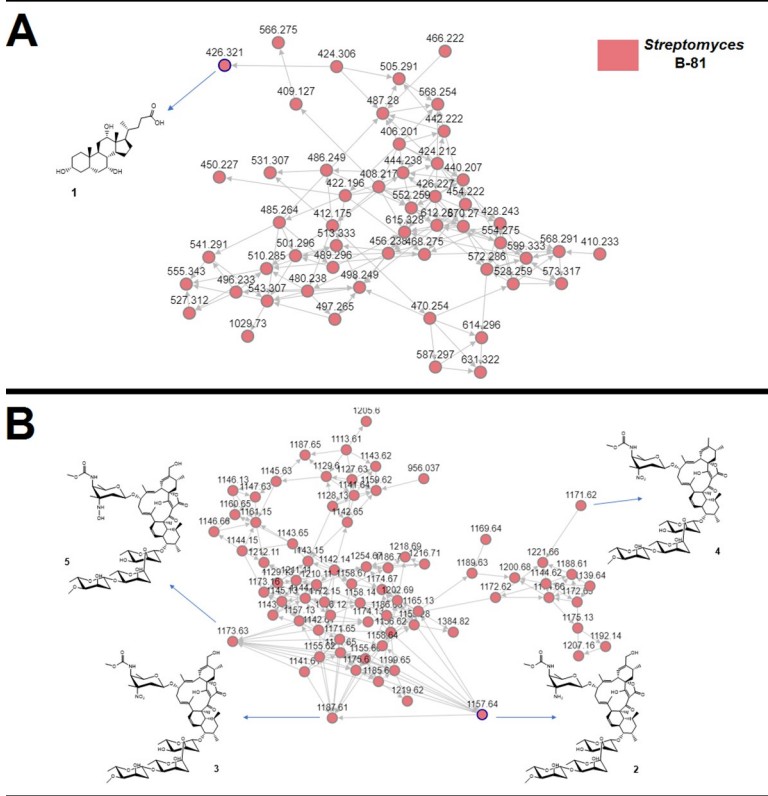

**Fig 3. Molecular networking obtained for *Streptomyces* sp. MW562807 extract Cholic Acid and its derivatives were grouped in cluster A, and Lobophorins were grouped in cluster B.** Nodes circled in blue indicate molecules identified by comparison with the GNPS platform database.

## Secondary metabolite analysis of *Streptomyces* sp. MW562807 isolate

The crude extract obtained from *Streptomyces* sp. MW562807 was analized by UHPLC-MS and demonstrated a broad biomolecule profile (S2 Fig). Furthermore, we screened the metabolites in the GNPS platform, and molecular networking revealed two clusters (A and B) that exhibited secondary metabolites produced by *Streptomyces* sp. MW562807 extract (pink) (Fig 3).

All compounds close to cholic acid are described in the pink circle (Fig 3), there are described as close compounds because they are grouped within the same cluster derived from the proximity of their specific masses and according to what it generates after comparing in the GNPS platform database. Thus, cluster B contains compounds called Loboforins with Loboforins A, B, E and K at number 2, 3, 4 and 5 with a specific masses of 1157.64, followed 1187.61, 1171.61, and 1173.63, respectively.

Metabolites were identified as a hit in the GNPS database or manually identified by accurate mass analyses, which showed mass errors below 5 ppm (Table 6).

The observed signals corresponded to Cholic Acid (1), Lobophorin A (2), Lobophorin B (3), Lobophorin E (4), and Lobophorin K (5).

In cluster A, the GNPS database indicated the presence of cholic acid in *Streptomyces* sp. MW562807 extract (S3 and S4 Figs). In molecular networking, each MS/MS spectrum is represented by nodes, which are grouped into clusters based on their fragmentation pattern similarity. Thus, compounds of the same molecule class are grouped in the same cluster [39]. The

**Table 6. MS data obtained for secondary metabolites detected in *Streptomyces* sp. MW562807 extract.**

| Compound | Ion formula | Calculated *m/z* | Experimental *m/z* | Error (ppm) |
|---|---|---|---|---|
| Cholic Acid | $C_{24}H_{44}NO_5$ | 426.3219 | 426.3212 | -1.6 |
| Lobophorin A | $C_{61}H_{93}N_2O_{19}$ | 1157.6372 | 1157.6373 | 0.1 |
| Lobophorin B | $C_{61}H_{91}N_2O_{21}$ | 1187.6114 | 1187.6108 | -0.5 |
| Lobophorin E | $C_{61}H_{91}N_2O_{20}$ | 1171.6165 | 1171.6160 | -0.4 |
| Lobophorin K | $C_{61}H_{93}N_2O_{20}$ | 1173.6322 | 1173.6321 | -0.1 |
| Compound 6 | $C_{11}H_{17}O_3$ | 197.1177 | 197.1173 | -2.0 |

Cholic Acid cluster is composed of various nodes, suggesting the presence of other analogs (Fig 4). Cholic acid and other bile acids have been reported as secondary metabolites of specific members of the genus *Streptomyces* and *Myroides* [57]. In the literature, Cholic Acid derivatives and bile acids are termed antimicrobial agents, displaying activity against Gram-positive and Gram-negative bacterial or improving the antimicrobial effect of antibiotics [58–61].

In cluster B, the GNPS database indicated the production of Lobophorin production by *Streptomyces* sp. MW562807 strain (S5 and S6 Figs), related to the loss of the sugar units in the structure, as revealed in literature [62]. Similar to cluster A, cluster B also revealed other Lobophorin compounds such as lobophorins B, E, and K, which were identified based on their accurate masses and fragmentation profiles with typical fragments at *m/z* 184.09, 157.09, 108.08, and 97.06 (S7–S9 Figs). Lobophorins were discovered by Jiang et al. (1999) in the expedition to Belize on board the Columbus research ship [63]. On the occasion, a new strain of actinomycete, named # CNC-837, was isolated from the surface of a brown algae from the Caribbean called *Lobophora variegata*, and was reported to produce lobophorins A and B, potent anti-inflammatory agents [63]. For the cytotoxic activity, Lobophorins A and E were inactive, while Lobophorins B and F were active against MCF-7 cells (human breast adenocarcinoma cell line), NCI-H460 (human non-small cell lung cancer cell line), and SF-268 (human glioma cell line) [64]. Further, Lobophorin C exhibited potent cytotoxic activity against human liver cancer cells, while Lobophorin D displayed a significant inhibitory effect on human breast cancer cells [65]. For the antibacterial activity, Lobophorins A, B, E, and F demonstrated activity against *Bacillus thuringensis* SCSIO BT01 [64], Lobophorin H exhibited antibacterial activity against *Bacillus subtilis* [66] and *S. olivaceus* SCSIO T05, a marine-derived strain, was previously reported to be isolated from the Indian Ocean deep-sea-derived sediment [67]. To explore the biosynthetic potential of this strain, metabolic engineering and

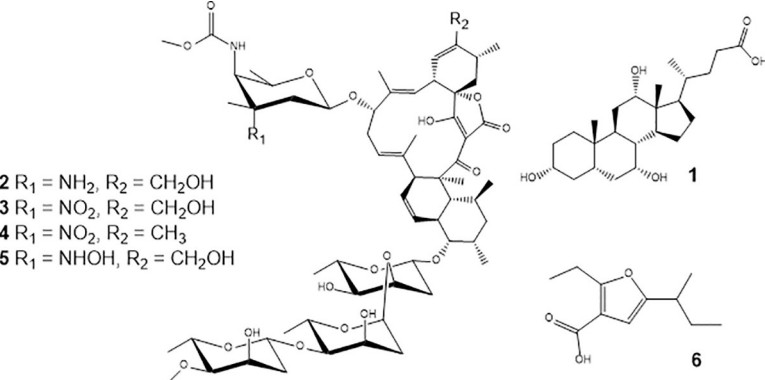

**Fig 4. Structures of secondary metabolites identified in *Streptomyces* sp. MW562807 extract.**

genome mining were performed in *S. olivaceus* SCSIO T05, leading to the isolation and identification of one known compound, Loboforin CR4 [68]. A compound of the Loboforin family, Lobophorin K promoted bacteriostatic effects against *Staphylococcus aureus* EPI1167 MSSA [69], *Streptomyces* sp. SCSIO 01127 was isolated from deep sea sediment in southern China and the phylogenetic tree generated by a neighbor-joining method clearly revealed that the evolutionary relationship had the greatest similarities to *Streptomyces olivaceus* NBRC 12805[T] (AB249920) (100%) and *Streptomyces pactum* NBRC 13433[T] (100%). Thus, this strain was designated as *Streptomyces* sp. SCSIO 01127 presented biological activities product of Lobophorins A, B, F and E against *Bacillus thuringensis* SCSIO BT01, *S. aureus* ATCC 29213, *E. faecalis* ATCC 29212 and three human tumor cell lines SF-268, MCF-7 and NCI—H460, thus the strain MW562807 that we present in this manuscript have active biological activities. [70].

Besides Cholic Acid and Lobophorins, a Furan-Type compound (6) was also detected in *Streptomyces* sp. MW562807 extract (S10 Fig). The structures of the metabolites **1**–**6** are shown in Fig 4.

Compound 6 exhibited fragments at m/z 168.07, 167.07, and 153.05, in agreement with the findings of a previous study [62]. Besides Lobophorin A, Compound 6 was reported as a secondary metabolite of the *Streptomyces* sp. VN1 strain, a microorganism isolated from sea sediment from the coastal region of Vietnam [62]. For the Compound 6 displayed by our data further investigation is needed to correlate the antimicrobial and cytotoxic activities, observed for *Streptomyces* sp. MW562807 extract, to Bile Acids, Lobophorins, and Furan Compounds. Additionally, extremophilic environments, such as Saloons of Bayovar, are prolific sources of microorganisms, including the identified unique *Streptomyces* sp. Due to the extreme conditions, these microorganisms produce different compounds to adapt to surviving in the environment, this makes *Streptomyces* sp. MW562807 an attractive source of bioactive compounds.

## Conclusion

This study successfully determined the diversity and bioactive potential of the actinobacterial isolates sourced from locations with extreme environments (Morrope and Bayovar saloons) in northwestern Peru; these regions had not been previously explored for this type of study. Furthermore, the Actinobacteria isolates, namely, *Streptomyces* sp. MW562814, MW562805, and MW562807, could be new species; their importance lies in their antibacterial and antiproliferative potential, with *Streptomyces* sp. MW562807 demonstrating the most promising activity. Six biomolecules (Cholic Acid; Lobophorin A, B, E, and K; in addition to a Sixth Compound) were detected in this cultivable actinobacterial isolated from Bayovar's saloons, but these have not yet been found in the GNPS curated database. In this study, we revealed that the Actinobacteria existing in these extreme environmental conditions have diverse characteristics, and could comprise new species that produce novel and biologically active compounds. The antibacterial and antiproliferative potential of these compounds renders the study of these saline lagoons of northwestern Peru valuable. In a Future work, the genome sequencing of these three Streptomyces species will be presented out to identify these molecules to purify and characterize them, as this can result in the economically beneficial production of bioactive compounds for future pharmaceutical applications.

## Supporting information

**S1 Fig. Flowchart depicting the methodological strategy adopted in this study.**
(DOCX)

**S2 Fig.** Total ion chromatogram (TIC) of UHPLC-MS analyses for (A) *Streptomyces* sp. MW562807 extract and (B) control.
(DOCX)

**S3 Fig.** MS/MS match between GNPS database (green) and Cholic Acid (1) from *Streptomyces* sp. MW562807 extract (black).
(DOCX)

**S4 Fig.** Extracted ion chromatograms of *m/z* 426.32 for (A) *Streptomyces* sp. MW562807 extract and (B) control. (C) Mass spectrum of ion $[M+NH_4]^+$ *m/z* 426.3212 obtained for compound Cholic Acid (**1**) (error = -1.6 ppm) at 8.1 min.
(DOCX)

**S5 Fig.** MS/MS match between GNPS database (green) and Lobophorin A (2) from *Streptomyces* sp. MW562807 extract (black).
(DOCX)

**S6 Fig.** Extracted ion chromatograms of *m/z* 1157.63 for (A) *Streptomyces* sp. MW562807 extract and (B) control. (C) Mass spectrum of ion $[M+H]^+$ *m/z* 1157.6373 obtained for Lobophorin A (**2**) (error = 0.1 ppm) at 7.9 min.
(DOCX)

**S7 Fig.** Extracted ion chromatograms of *m/z* 1187.61 for (A) *Streptomyces* sp. MW562807 extract and (B) control. (C) Mass spectrum of ion $[M+H]^+$ *m/z* 1187.6108 obtained for Lobophorin B (**3**) (error = -0.5 ppm) at 6.9 min. (D) MS/MS spectrum of Lobophorin B.
(DOCX)

**S8 Fig.** Extracted ion chromatograms of *m/z* 1171.61 for (A) *Streptomyces* sp. MW562807 extract and (B) control. (C) Mass spectrum of ion $[M+H]^+$ *m/z* 1171.6160 obtained for Lobophorin E (**4**) (error = -0.4 ppm) at 7.2 min. (D) MS/MS spectrum of Lobophorin E.
(DOCX)

**S9 Fig.** Extracted ion chromatograms of *m/z* 1173.63 for (A) *Streptomyces* sp. MW562807 extract and (B) control. (C) Mass spectrum of ion $[M+H]^+$ *m/z* 1173.6321 obtained for Lobophorin K (**5**) (error = -0.4 ppm) at 7.2 min. (D) MS/MS spectrum of Lobophorin K.
(DOCX)

**S10 Fig.** Extracted ion chromatograms of *m/z* 197.11 for (A) *Streptomyces* sp. MW562807 extract and (B) control. (C) Mass spectrum of ion $[M+H]^+$ *m/z* 197.1173 obtained for compound **6** (error = -2.0 ppm) at 7.2 min. (D) MS/MS spectrum of Compound **6**.
(DOCX)

## Acknowledgments

We are grateful to the anonymous reviewers whose constructive criticism has significantly improved the quality of the manuscript, and we would like to thank Editage (www.editage. com) for English language editing.

## Author Contributions

**Investigation:** Rene Flores Clavo, Nataly Ruiz Quiñones, Luis Miguel Serquen Lopez, Fabiana Fantinatti Garboggini.

**Methodology:** Álvaro Jose Hernández-Tasco, Marcos José Salvador, Ana Lúcia Tasca Gois
Ruiz, Lúcia Elaine de Oliveira Braga, Jonas Henrique Costa, Taícia Pacheco Fill, Zhandra
Lizeth Arce Gil.

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
