## [Decision Letter · Decision Letter 0]

23 Nov 2020

PONE-D-20-29463

EVALUATION OF ANTIMICROBIAL AND ANTIPROLIFERATIVE ACTIVITIES OF ACTINOBACTERIA ISOLATED FROM THE SALINE LAGOONS OF NORTHWEST PERU

PLOS ONE

Dear Dr. Flores Clavo,

Thank you for submitting your manuscript to PLOS ONE. After careful consideration, we feel that it has merit but does not fully meet PLOS ONE’s publication criteria as it currently stands. Therefore, we invite you to submit a revised version of the manuscript that addresses the points raised during the review process.

The comments by both referees are included at the end of this letter and I would kindly suggest you follow them should you decide to resubmit your MS to PLOS ONE. Both referees concluded -separately- that your MS falls under the "Major revision" decision and I also agree with them completely. Despite the particular comments by each reviewer I would kindly suggest that a major revision of the written English is performed since this will also increase the impact of the MS.   

We look forward to receiving your revised manuscript.

Kind regards,

Luis Angel Maldonado Manjarrez, Ph.D.

Academic Editor

PLOS ONE

Journal Requirements:

2. We note that Figure 1 in your submission contains map images which may be copyrighted. All PLOS content is published under the Creative Commons Attribution License (CC BY 4.0), which means that the manuscript, images, and Supporting Information files will be freely available online, and any third party is permitted to access, download, copy, distribute, and use these materials in any way, even commercially, with proper attribution. For these reasons, we cannot publish previously copyrighted maps or satellite images created using proprietary data, such as Google software (Google Maps, Street View, and Earth). For more information, see our copyright guidelines: http://journals.plos.org/plosone/s/licenses-and-copyright.

(1) You may seek permission from the original copyright holder of Figure 1 to publish the content specifically under the CC BY 4.0 license. 

3. Please ensure that reagents are described in sufficient detail for another researcher to reproduce the experiments described (source, product number, lot number).

4. Please provide additional information about each of the cell lines used in this work, including any quality control testing procedures (authentication, characterisation, and mycoplasma testing). For more information, please see http://journals.plos.org/plosone/s/submission-guidelines#loc-cell-lines.

5. To comply with PLOS ONE submission guidelines, in your Methods section, please provide additional information regarding your statistical analyses. For more information on PLOS ONE's expectations for statistical reporting, please see https://journals.plos.org/plosone/s/submission-guidelines.#loc-statistical-reporting.

6. We suggest you thoroughly copyedit your manuscript for language usage, spelling, and grammar. If you do not know anyone who can help you do this, you may wish to consider employing a professional scientific editing service.  

7. Thank you for stating the following in the Acknowledgments Section of your manuscript:

"At the research agency of Peru (Concytec-Fondecyt) financially supported this work within the framework of the 041-01 call with the 190-2018 contract; as well as the National Council for Scientific and Technological Development of Brazil (CNPq), it was the authors are grateful to the cooperation of the postgraduate programs in Genetics and Molecular Biology and Biosciences of the Biology Institute of the State University of Campinas. Also, Coordination for the Improvement of Higher Education Personnel - Brazil (CAPES) - (Scholarship-code 001)."

"CONFLICT OF INTEREST STATEMENT

The authors declare no competing financial interests and the funders had no role in study design, data collection and analysis, decision to publish, or preparation of the manuscript."

Reviewers' comments:

Reviewer's Responses to Questions

**Comments to the Author**

1. Is the manuscript technically sound, and do the data support the conclusions?

Reviewer #1: Yes

Reviewer #2: Partly

2. Has the statistical analysis been performed appropriately and rigorously? 

Reviewer #1: Yes

Reviewer #2: No

3. Have the authors made all data underlying the findings in their manuscript fully available?

Reviewer #1: No

Reviewer #2: No

4. Is the manuscript presented in an intelligible fashion and written in standard English?

Reviewer #1: No

Reviewer #2: No

5. Review Comments to the Author

Reviewer #1: This manuscript describes the isolation of actinobacterial strains from a saline lagoon in Northwest Peru. The authors identified a selected group of actinobacteria by 16S rRNA gene sequencing and subjected the isolates to tests in order to determine whether they produce any bio-active compounds with activity against bacteria and various cancer cell lines. The extract from one of the most promising strains was evaluated via molecular networking, allowing for the prediction of compounds produced by this strain.

It is clear that the manuscript represents a large body of work, which also resulted in a rather lengthy manuscript. The authors would need to look at ways in which the manuscript can be shortened so that it is more streamlined. In addition, the authors also need to make sure that the way of citing references in the main text follow the journal’s requirements.

Here are some additional comments/suggestions to the authors:

1. The manuscript will require language editing before it can be considered for publication.

2. Figure 1: the caption for image ‘d’ is not included in the figure legend.

3. Figure 2 can probably be presented as supplementary material.

4. Page 8, section 2.2.: The authors described a very specific isolation protocol. This obviously would limit the amount and type of actinobacteria that would be isolated. What was the rationale behind using this method and were other isolation methods explored, but yielded no isolates of interest? In addition, many researchers often base their isolation protocol on the observed physicochemical characteristics of the environment the sample was collected from. Were any physicochemical analyses performed (e.g. sediment pH, level of salinity, presence of K, P, N, C, etc.)?

In addition, in this section and throughout the manuscript, when indicating a % solution, please indicate whether it is v/v or w/v, e.g. isolates were preserved in 20% (v/v) glycerol…

5. Page 9, section 2.3.1: What was the final pH of the R2A-ASW broth used?

6. Page 9-10, section 2.32: It is mentioned that modifications were made to the DNA extraction method reported by Pospiech and Neuman (1995), but no details are provided. Please expand on the modifications effected.

Line 161: The PCR protocol also calls for a large amount of DNA to be added. Is there any particular reason for that?

Line 166: What % agarose gel was used for the analysis of the amplicons?

Lines 171-172: Which program was used to assemble to DNA sequences?

7. Page 11, line 186: What was the rationale behind the use of R2A broth for the production of bio-active compounds? Actinobacteria can produce numerous different compounds, a process that is greatly affected by the type of growth media used. Were different types of media tested, or did you solely focus on this medium?

8. Page 11, lines 196-197: Here mention is made that the crude extracts were also analysed by LC-MS. Is this the same analyses reported later on in the methodology section (in which case the authors need to refer to the section) or was this a separate analysis (in which case more details are required)?

9. Page 12, line 208: The wording of this sentence does not make sense – the ‘three crude bacterial extracts’ can’t be ‘tested as antimicrobial producers’. Best to say that they were tested for antimicrobial activity.

10. Page 12, lines 211-212: What is meant by ‘pre-coated microbial cultures’?

11. Page 12, lines 217-218: Define the abbreviations ‘MDRs’ and ‘UCI’ – these have not been defined previously.

12. Throughout the manuscript, make sure that ‘sp.’ is not written in italics, e.g. page 13, line 244 – the ‘sp.’ should not be in italics.

13. Page 13, lines 244-245: It is indicated that the B-81 extract was diluted 1 in 10 in methanol, but there is no indication as to the final concentration of the extract. Please indicate, e.g. 1mg/ml, 0.1mg/ml, etc.

14. Page 15, section 3.1: The use of phenotypic characteristics to group actinobacteria is notoriously unreliable. Genera within a family may have very similar morphological features and you stand a chance of ‘missing out’ potentially unique/’rare’ actinobacteria, e.g. members of Kitasatospora and Streptomyces share phenotypic features, similarly members of Pseudonocardia and Nonomuraea. Why not identify all 166 strains?

15. Page 17: The text provided in this section is basically a repeat of what is presented in Table 3. Only highlight the most interesting rather than repeating what is already presented.

16. Page 18, line 317 and 324: ‘Streptomyces’ should be written in italics.

17. Page 18, lines 321-325: Here the authors highlight the types of actinobacteria isolated from the different environments. Care should be taken here especially since the authors only made use of a single isolation approach, which will result in isolation bias. They therefore can’t compare what they detected during their isolations to that of others – they can simply report it and mention that these were isolated under the experimental conditions applied.

18. Figure 3: A phylogenetic tree containing only the three selected strains (and related streptomycetes) is presented in this figure. It seems a bit out of place since the importance of these three strains only becomes clear once the results for antimicrobial and anticancer tests have been reported. This makes it seem out of order. I would recommend amending the tree to include all the streptomycete strains identified in the study and once the bio-activity has been reported, reference can be made to this figure. This also means that Table 3 can be shifted to supplementary material.

19. Page 20, line 364: No evidence has been provided to support the classification of the isolates as ‘extremophilic’.

20. Page 22, line 395: Note the spelling of ‘doxorubicin’ – please correct.

21. Page 22, line 398: It is mentioned that the extracts from ‘three different media’ were selected. This is not reported in the methodology section – please amend.

22. Page 30, line 544: Note the spelling of ‘cholic’ – please correct.

23. Page 18, line 333: incomplete surname provided of the reference Cortés-Albayay.

24. Page 36, reference 40: Not cited in the main text.

25. Page 37, reference 42: Not cited in main text or incorrect spelling of lead author.

26. Page 37, reference 48: Not cited in main text or incorrect spelling of l

Reviewer #2: The article “EVALUATION OF ANTIMICROBIAL AND ANTIPROLIFERATIVE ACTIVITIES OF ACTINOBACTERIA ISOLATED FROM THE SALINE LAGOONS OF NORTHWEST PERU” describes the isolation and antimicrobial and anticancer capacity of strains isolated from saline environments. Although some of the results are of interest the quality of the manuscript is not enough, and a careful revision of data presentation and English should be taken into account before publication.

It is not clear at any point, why the three strains of Streptomyces have been selected for further analysis, it is said that they have better inhibition results, but it has not been shown any experiment with the other strains.

The authors should clarify that the sampling was in agreement with Nagoya protocol (check comment from line 110).

Lines 30-31: Reconsider this sentence, it has no many sense as it is.

Line 35: add “and /or”

Line 36: add “these bacteria” before “were identified”

Lines 36-38: I imagine these numbers refer to abundance of strains but it is not clear here.

Line 38: Actinobacteria should be in italics

Line 60: Here should be a final dot instead of semicolon.

Line 69: Italics

Line 77: give a reference.

Line 110: The Nagoya protocol was signed by Peru in 2014 so, at least the sampling of 2015 should be authorised by the goverment.

Line 112: it should be b, c and d, isn't it?

Table 1: before was indicated that sampling was in July, clarified it. Indeed if it was in december the Nagoya protocol should be applied.

Figure 2: correct “bacterial” and “extremophilic bacteria”. But indeed this figure can be omitted.

Line 126: it should be actinomycetes, not actinomyces, but indeed better if you use "Actinobacteria"

Line 144: this amount means 2.43% not 5%, verified your data.

Lines 276-279: These characteristics do not allow the classification into genus, you can group them into groups with similar characteristics, but not genus. This classification could be obtained after 16S rRNA gene analysis.

Line 286: based on what? which strains were included in those groups? this should be further explained.

Line 288: What are the other 116 strains isolated?

Page 292: delete strain, and put "isolates" and “samples” in plural

Line 303: correct to “were similar” or “presented higher similarities with”

Lines 300-317: it has no sense to give all the data here again if you give them in table 3, keep just the text or the table but not both. I recommend to keep the table.

Line 317: Streptomyces should be in italics; delete “of the”

Line 318: change “to” by “from”

Line 323: add a space before “has”

Lines 324 and 388: Streptomyces should be in italics

Line 328: correct the reference

Figure 3: I do not understand why the authors has selected only three of their strains to construct the phylogenetic tree, a good analysis of their strains should include all the ones for which the 16s rRNA gene has been obtained, independently of their ability to produce secondary metabolites. The 16S rRNA gene sequences should be deposited in public databases, and their accession numbers included in the tree.

Line 345: genus names should be in italics.

Line 353-356: this is not true, that is only true for the strain M-92, but not for the other two strains represented in the tree.

Line 379: check the reference

Line 401: delete “the”

Lines 405-407: these values are higher than 50, Shouldn’t they be considered as inactive according to line 394 reference? This should be further discussed by the authors.

Line 474: give details about the molecules marked as 1, 2 ,3, 4, and 5 in the figure.

Line 541: it should be a comma after the first “study”. Indeed one “study” should be replaced.

Line 542: correct “actinobacteria”

Line 550: delete the final dot after “potential”

6. PLOS authors have the option to publish the peer review history of their article (what does this mean?). If published, this will include your full peer review and any attached files.

Reviewer #1: No

Reviewer #2: No

---

## [Author Response · Author response to Decision Letter 0]

3 Mar 2021

Response to the Editor comments

Comments from editor:

Reply: We have corrected the tittle, headings, authors’ names, and affiliations according to PLOS ONE's style requirements.

2. We note that figure 1 in your submission contains map images which may be copyrighted. All PLOS content is published under the Creative Commons Attribution License (CC BY 4.0), which means that the manuscript, images, and Supporting Information files will be freely available online, and any third party is permitted to access, download, copy, distribute, and use these materials in any way, even commercially, with proper attribution. For these reasons, we cannot publish previously copyrighted maps or satellite images created using proprietary data, such as Google software (Google Maps, Street View, and Earth). For more information, see our copyright guidelines.

Reply: We have modified figure 1 in our manuscript. Now, the map has been removed and only the sampling sites photos remain.

3. Please ensure that reagents are described in sufficient detail for another researcher to reproduce the experiments described (source, product number, lot number).

Reply: We have made elaborated detailed protocols which are now published in protocols.io to make the requested information available. 

4. Please provide additional information about each of the cell lines used in this work, including any quality control testing procedures (authentication, characterization, and mycoplasma testing).

Reply: We have added the requested information about the cell lines in the description to methods.

5. To comply with PLOS ONE submission guidelines, in your Methods section, please provide additional information regarding your statistical analyses.

Reply: We have added detailed information in the 2.7 Statistical analysis in the Methods section.

6. We suggest you thoroughly copyedit your manuscript for language usage, spelling, and grammar.

Reply: We have purchased the services of the suggested Editage editors for the language corrections.

"CONFLICT OF INTEREST STATEMENT

The authors declare no competing financial interests and the funders had no role in study design, data collection and analysis, decision to publish, or preparation of the manuscript."

Reply: We have removed the funding information from the acknowledgments section and put it on the Funding Statement in the Cover Letter.

8. Please include captions for your Supporting Information files at the end of your manuscript, and update any in-text citations to match accordingly.

Reply: We have added the captions information in the Supporting information section.

Comments from the nomenclature reviewers to be forwarded to the authors: The nomenclature reviewers agree with the proposed name and its etymology.

Reply: Thank you.

Response to the reviewer’s comments

Reviewer #1: This manuscript describes the isolation of actinobacterial strains from a saline lagoon in Northwest Peru. The authors identified a selected group of actinobacteria by 16S rRNA gene sequencing and subjected the isolates to tests in order to determine whether they produce any bio-active compounds with activity against bacteria and various cancer cell lines. The extract from one of the most promising strains was evaluated via molecular networking, allowing for the prediction of compounds produced by this strain. 

It is clear that the manuscript represents a large body of work, which also resulted in a rather lengthy manuscript. The authors would need to look at ways in which the manuscript can be shortened so that it is more streamlined. In addition, the authors also need to make sure that the way of citing references in the main text follow the journal’s requirements.

Here are some additional comments/suggestions to the authors:

1. The manuscript will require language editing before it can be considered for publication.

REPLY: According to your suggestions, this manuscript has been sent to Editage Editors for language corrections.

2. Figure 1: the caption for image 'd' is not included in the figure legend.

REPLY: We have modified figure 1 in our manuscript. Now, the map has been removed and only the sampling sites photos remain. Now “d” is included in Figure 1.

3. Figure 2 can probably be presented as supplementary material.

REPLY: We have put Figure 2 to the Supplementary material section.

4. Page 8, section 2.2.: The authors described a very specific isolation protocol. This obviously would limit the amount and type of actinobacteria that would be isolated. What was the rationale behind using this method and were other isolation methods explored, but yielded no isolates of interest? In addition, many researchers often base their isolation protocol on the observed physicochemical characteristics of the environment the sample was collected from. Were any physicochemical analyses performed (e.g., sediment pH, level of salinity, presence of K, P, N, C, etc.)?

REPLY: We have developed and optimized a selective isolation protocol because the purpose of the research was to recover only actinobacteria, at the expense of Bacillus and Micrococcus, which has a faster growth. We have considered pH, solute concentrations, and other conditions from previous physicochemical analysis. 

In addition, in this section and throughout the manuscript, when indicating a % solution, please indicate whether it is v/v or w/v, e.g., isolates were preserved in 20% (v/v) glycerol...

REPLY: Thank you. We have corrected those specifications about percentage.

5. Page 9, section 2.3.1: What was the final pH of the R2A-ASW broth used?

REPLY: We have corrected that specification o pH, which is 7.

6. Page 9-10, section 2.32: It is mentioned that modifications were made to the DNA extraction method reported by Pospiech and Neuman (1995), but no details are provided. Please expand on the modifications effected.

Line 161: The PCR protocol also calls for a large amount of DNA to be added. Is there any reason for that?

Reply: We have made elaborated detailed protocols which are now published in protocols.io to make the requested information available. Link: https://dx.doi.org/10.17504/protocols.io.bpvjmn4n

Line 166: What % agarose gel was used for the analysis of the amplicons?

Reply: We have made elaborated detailed protocols which are now published in protocols.io to make the requested information available. Link: https://dx.doi.org/10.17504/protocols.io.brrmm546

Lines 171-172: Which program was used to assemble to DNA sequences?

REPLY: The BioEdit 7.0 software was used for DNA sequences assembly and it is now written in the Methods section.

7. Page 11, line 186: What was the rationale behind the use of R2A broth for the production of bio-active compounds? Actinobacteria can produce numerous different compounds, a process that is greatly affected by the type of growth media used. Were different types of media tested, or did you solely focus on this medium?

REPLY: We have made elaborated detailed protocols which are now published in protocols.io to make the requested information available. R2A Broth, Nutrient Broth and SP2 Broth.

8. Page 11, lines 196-197: Here mention is made that the crude extracts were also analysed by LC-MS. Is this the same analyses reported later on in the methodology section (in which case the authors need to refer to the section) or was this a separate analysis (in which case more details are required)?

REPLY: Those analysis are the same. We have now corrected the reference in the Methods section.

9. Page 12, line 208: The wording of this sentence does not make sense – the 'three crude bacterial extracts' can't be 'tested as antimicrobial producers'. Best to say that they were tested for antimicrobial activity.

Reply: We have corrected that sentence and now it says “tested for antimicrobial activity.

10. Page 12, lines 211-212: What is meant by 'pre-coated microbial cultures'?

REPLY: We have corrected that sentence in the Methods section and now it just says that they were added.

11. Page 12, lines 217-218: Define the abbreviations 'MDRs' and 'UCI' – these have not been defined previously.

REPLY: We have detailed the definitions of those acronyms in the Methods section.

12. Throughout the manuscript, make sure that 'sp.' is not written in italics, e.g. page 13, line 244 – the 'sp.' should not be in italics.

REPLY: We have now corrected the italics in the “sp” throughout the manuscript. 

13. Page 13, lines 244-245: It is indicated that the MW562807 extract was diluted 1 in 10 in methanol, but there is no indication as to the final concentration of the extract. Please indicate, e.g. 1mg/ml, 0.1mg/ml, etc.

REPLY: We have now specified the final concentration of 6.4 mg/mL of the MW562807 extract in the Methods section. 

14. Page 15, section 3.1: The use of phenotypic characteristics to group actinobacteria is notoriously unreliable. Genera within a family may have very similar morphological features and you stand a chance of 'missing out' potentially unique/'rare' actinobacteria, e.g. members of Kitasatospora and Streptomyces share phenotypic features, similarly members of Pseudonocardia and Nonomuraea. Why not identify all 166 strains?

REPLY: We did not identify all 166 strains because we were identifying bacteria because the main objective was to know the bacteria with antimicrobial and antiproliferative activities.

15. Page 17: The text provided in this section is basically a repeat of what is presented in Table 3. Only highlight the most interesting rather than repeating what is already presented.

REPLY: We have now kept only the most interesting instead of than repeating the same as Table 3.

16. Page 18, line 317 and 324: 'Streptomyces' should be written in italics.

REPLY: We have now corrected that word.

17. Page 18, lines 321-325: Here the authors highlight the types of actinobacteria isolated from the different environments. Care should be taken here especially since the authors only made use of a single isolation approach, which will result in isolation bias. They therefore can't compare what they detected during their isolations to that of others – they can simply report it and mention that these were isolated under the experimental conditions applied.

REPLY: We have now specified that we just report those bacteria according to the conditions of the samples they were obtained.

18. Figure 3: A phylogenetic tree containing only the three selected strains (and related streptomycetes) is presented in this figure. It seems a bit out of place since the importance of these three strains only becomes clear once the results for antimicrobial and anticancer tests have been reported. This makes it seem out of order. I would recommend amending the tree to include all the Streptomycete strains identified in the study and once the bio-activity has been reported, reference can be made to this figure. This also means that Table 3 can be shifted to supplementary material.

REPLY: The phylogenetic tree is has been edited considering a representative strains for each of the 13 identified sub-groups.

19. Page 20, line 364: No evidence has been provided to support the classification of the isolates as 'extremophilic'.

REPLY: We have now specified that they were considered as extremophilic since they are able to grow in all media (Artificial Sea Water - ASW) supplemented with 5% of NaCl. 

20. Page 22, line 395: Note the spelling of 'doxorubicin' – please correct.

REPLY: Thank you. We have now corrected that spelling.

21. Page 22, line 398: It is mentioned that the extracts from 'three different media' were selected. This is not reported in the methodology section – please amend.

REPLY: We have now specified that we just report those bacteria according to the conditions of the samples they were obtained.

22. Page 30, line 544: Note the spelling of 'cholic' – please correct.

REPLY: Thank you. We have now corrected that word. 

23. Page 18, line 333: incomplete surname provided of the reference Cortés-Albayay.

REPLY: Thank you. We have now corrected that reference. Now we are using Mendeley to manage references.

24. Page 36, reference 40: Not cited in the main text.

REPLY: Thank you. We have now corrected that citation. Now we are using Mendeley to manage references.

25. Page 37, reference 42: Not cited in main text or incorrect spelling of lead author.

REPLY: Thank you. We have now corrected that citation. Now we are using Mendeley to manage references.

26. Page 37, reference 48: Not cited in main text or incorrect spelling

REPLY: Thank you. We have now corrected that citation. Now we are using Mendeley to manage references.

Reviewer #2: The article "EVALUATION OF ANTIMICROBIAL AND ANTIPROLIFERATIVE ACTIVITIES OF ACTINOBACTERIA ISOLATED FROM THE SALINE LAGOONS OF NORTHWEST PERU" describes the isolation and antimicrobial and anticancer capacity of strains isolated from saline environments. Although some of the results are of interest the quality of the manuscript is not enough, and a careful revision of data presentation and English should be taken into account before publication. It is not clear at any point, why the three strains of Streptomyces have been selected for further analysis, it is said that they have better inhibition results, but it has not been shown any experiment with the other strains. The authors should clarify that the sampling was in agreement with Nagoya protocol (check comment from line 110).

REPLY: Thank you. We have now specified that sampling was carried out according to The Nagoya protocol in the Methods section. 

Lines 30-31: Reconsider this sentence, it has no many sense as it is.

REPLY: Thank you. We have now corrected those lines.

Line 35: add "and /or"

REPLY: Thank you. We have added “and/or”.

Line 36: add "these bacteria" before "were identified"

REPLY: Thank you. We have added “these bacteria”

Lines 36-38: I imagine these numbers refer to abundance of strains but it is not clear here.

REPLY: Thank you. We have now edited that line in the Methods section. We now just mention the Streptomyces genera without expressing quantities to avoid confusions.

Line 38: Actinobacteria should be in italics

REPLY: Thank you. We have now corrected that word.

Line 60: Here should be a final dot instead of semicolon.

REPLY: Thank you. We have now corrected the punctuation.

Line 69: Italics

REPLY: We have now corrected the italics.

Line 77: give a reference.

REPLY: We have now provided the corresponding reference.

Line 110: The Nagoya protocol was signed by Peru in 2014 so, at least the sampling of 2015 should be authorised by the goverment.

REPLY: We have initiated the corresponding procedures according to Peruvian laws we are currently obtaining the final documentation for the samplings from 2015 since the competent Governmental Entities and Normative Guidelines for the access to genetic resources specific to these sampling environments have been stablished recently in 2020, the print of the message is sent to the responsible entity with whom you are working to obtain the agreement for the use of the genetic resources. 

Line 112: it should be b, c and d, isn't it?

REPLY: Yes, thank you. We have now modified Figure 1.

Table 1: before was indicated that sampling was in July, clarified it. Indeed if it was in december the Nagoya protocol should be applied.

REPLY: Thank you. The procedures according to the Nagoya protocol were stablished in 2020 for the specific conditions and samples sites. We are currently obtaining the final documentation for the access to genetic resources.

Figure 2: correct "bacterial" and "extremophilic bacteria". But indeed this figure can be omitted.

REPLY: Thank you. We have now corrected in supplementary Fig 1.”.

Line 126: it should be actinomycetes, not actinomyces, but indeed better if you use "Actinobacteria"

REPLY: We have now corrected that word.

Line 144: this amount means 2.43% not 5%, verified your data.

REPLY: We have now specified that ASW was supplemented with 5% of NaCl described protocol with reference 23.

Lines 276-279: These characteristics do not allow the classification into genus, you can group them into groups with similar characteristics, but not genus. This classification could be obtained after 16S rRNA gene analysis.

REPLY: We have modified that part in the Methods section and we classify them grouped in bacterial with similar characteristics.

Line 286: based on what? which strains were included in those groups? this should be further explained.

REPLY: We based this grouping on their genotypic similarities after the 16s rDNA gene alignment as shown in Table 2, and figure 2.

Line 288: What are the other 116 strains isolated?

REPLY: We isolated a total amount of 166 strains and then we could re-activate only 50 strains that were grouped into 13 clusters of actinobacteria identified by 16S rRNA. 

Page 292: delete strain, and put "isolates" and "samples" in plural

REPLY: Thank you. We have now corrected that in the Results section.

Line 303: correct to "were similar" or "presented higher similarities with"

REPLY: Thank you. We have corrected that line.

Lines 300-317: it has no sense to give all the data here again if you give them in table 3, keep just the text or the table but not both. I recommend to keep the table.

REPLY: Thank you. We have now deleted in the table.

Line 317: Streptomyces should be in italics; delete "of the"

REPLY: Thank you. We have edited that word and that line.

Line 318: change "to" by "from"

REPLY: Thank you. We have edited that word.

Line 323: add a space before "has"

REPLY: Thank you. We have edited that word.

Lines 324 and 388: Streptomyces should be in italics

REPLY: Thank you. We have edited that word.

Line 328: correct the reference

REPLY: Thank you. We have edited that reference.

Figure 3: I do not understand why the authors has selected only three of their strains to construct the phylogenetic tree, a good analysis of their strains should include all the ones for which the 16s rRNA gene has been obtained, independently of their ability to produce secondary metabolites. The 16S rRNA gene sequences should be deposited in public databases, and their accession numbers included in the tree.

REPLY: Thank you. We made the phylogenetic tree considering the three. Representative strains from each group from 13 sequenced strains. Also, we have now added information about de deposited sequences in NCBI database.

Line 345: genus names should be in italics.

REPLY: Thank you. We have edited that word.

Line 353-356: this is not true, that is only true for the strain M-92, but not for the other two strains represented in the tree.

REPLY: Thank you. We have now corrected that information in the Results section.

Line 379: check the reference

REPLY: Thank you. We have now corrected the reference.

Line 401: delete "the"

REPLY: Thank you. We have deleted the “the”.

Lines 405-407: these values are higher than 50, Shouldn't they be considered as inactive according to line 394 reference? This should be further discussed by the authors.

REPLY: Thank you. We have corrected and specified that information in the Results section.

Line 474: give details about the molecules marked as 1, 2 ,3, 4, and 5 in the figure.

REPLY: Thank you. We have now made a description specifying those molecules in the figure.

Line 541: it should be a comma after the first "study". Indeed one "study" should be replaced.

REPLY: Thank you. We have edited that line.

Line 542: correct "actinobacteria"

REPLY: Thank you. We have edited that word.

Line 550: delete the final dot after "potential"

REPLY: Thank you. We have now corrected the punctuation.

---

## [Decision Letter · Decision Letter 1]

27 May 2021

PONE-D-20-29463R1

EVALUATION OF ANTIMICROBIAL AND ANTIPROLIFERATIVE ACTIVITIES OF ACTINOBACTERIA ISOLATED FROM THE SALINE LAGOONS OF NORTHWEST PERU

PLOS ONE

Dear Dr. Flores Clavo,

Thank you for submitting your manuscript to PLOS ONE. After careful consideration, we feel that it has merit but does not fully meet PLOS ONE’s publication criteria as it currently stands. Therefore, we invite you to submit a revised version of the manuscript that addresses the points raised during the review process.

As you can notice from the comments of the two reviewers, one has suggested major and one has suggested minor revisions to your manuscript. However, I have decided to go for the "minor revisions" decision based on the first revision of your manuscript. Despite the latter, I would kindly appreciate if you would check on those comments of both reviewers and address all of their points accordingly as indicated in the previous paragraph. 

We look forward to receiving your revised manuscript.

Kind regards,

Luis Angel Maldonado Manjarrez, Ph.D.

Academic Editor

PLOS ONE

Journal Requirements:

Reviewers' comments:

Reviewer's Responses to Questions

**Comments to the Author**

1. If the authors have adequately addressed your comments raised in a previous round of review and you feel that this manuscript is now acceptable for publication, you may indicate that here to bypass the “Comments to the Author” section, enter your conflict of interest statement in the “Confidential to Editor” section, and submit your "Accept" recommendation.

Reviewer #1: (No Response)

Reviewer #3: (No Response)

2. Is the manuscript technically sound, and do the data support the conclusions?

Reviewer #1: Yes

Reviewer #3: Yes

3. Has the statistical analysis been performed appropriately and rigorously? 

Reviewer #1: Yes

Reviewer #3: N/A

4. Have the authors made all data underlying the findings in their manuscript fully available?

Reviewer #1: Yes

Reviewer #3: Yes

5. Is the manuscript presented in an intelligible fashion and written in standard English?

Reviewer #1: Yes

Reviewer #3: Yes

6. Review Comments to the Author

Reviewer #1: It is clear from the revised manuscript that the authors applied great effort to address the comments/recommendations/queries of the reviewers. However, with the editing and additions there are still some additional comments/recommendations for the revised manuscript:

1. Page 2-3, lines 54-55: ‘Three *Streptomyces* sp. strains’ sound cumbersome – consider changing the wording to ‘Three streptomycetes, strains MW562814,…’

2. Page 3, lines 59-60: To streamline the sentence, consider writing ‘… and the multi-drug resistant *Acinetobacter baumannii* AC-972.’

3. Page 3, line 61: ‘…in the R2A medium was determined using a…’

4. Page 4, lines 79-81: Consider merging the two sentences to provide more flow to the information provided, e.g., ‘…were reported from saline soils of the ephemeral salty lakes in Buryatiya [4], while *Streptomyces*, *Nocardiopsis*, and…’

5. Page 4, lines 81-83. The sentence on biological activities seems out of place – I would recommend removing it.

6. Page 4, line 89: ‘…to the production of enzymes, which can…’

7. Page 8, lines 147, 156, 159-160: It is conventional to refer to 16S rRNA (as per other sections in the manuscript).

8. Page 8, line 158: What is meant by ‘represented’? Did you mean that the sequences were submitted to the EZBioCloud 16S database ‘Identify’ service?

9. Page 9, line 162: ‘Mega’ should be written as ‘MEGA’

10. Throughout the manuscript: For any % solutions, please indicate whether they are v/v or w/v solutions.

11. Page 10, line 187: What is meant by ‘adjusted to a total volume of 200�l’? And is the ‘plate’ referred to a 96-well plate?

12. Page 13, lines 251-252: ‘…were used to group isolates into 42 filamentous and 8 non-filamentous bacterial groups’. Can’t use the information to ‘classify’, rather ‘group’, and since all the isolates were bacteria, best to keep it generic at this stage and use an inclusive term such as ‘bacteria’.

13. Table 2: The Family names should also be written in italics.

14. Page 14, line 283: Here is it indicated that MEGA7 was used, whereas in the methodology section it is indicated that MEGA6 was used – please correct.

15. Page 15, line 298: Remove ‘to’

16. Page 15, line 303: Replace ‘Actinobacteria’ with ‘actinobacterium’

17. Page 16, line 310: Replace ‘like’ (colloquial term) with ‘such as’ (more scientific).

18. Page 16, line 312: Did you mean to say that a member of the genus *Pseudonocardia* was also recovered during this study? In which case, the wording needs to change to reflect this: ‘…and plants, was also recovered in this study.’

19. Page 16, line 316 and throughout the manuscript: If there is no reference made to specific strains of a genus, then it is best to write out spp. as species. For example, here, it is best to write ‘…*Streptomyces* species as the most…’

20. Table 3: Replace ‘Isolates of *Streptomyces* sp.’ with ‘*Streptomyces* isolates’

21. Figure 3: It is recommended that the figure should be removed. The addition of all streptomycete isolates to Figure 2 negated the need for this figure.

22. Page 22: Be consistent in how the names of bioactive compounds are presented, e.g., shellmicin is written in both lower case and upper case in this section (see lines 430 and 436 vs 434 and 436).

23. Figure 4: How does the inset information (*Streptomyces* B-81 and Control) relate to the figure presented?

24. Page 25, line 463: What is meant by ‘rolling in the GNPS platform database’?

25. Page 25: Please see comment above about writing the names of bioactive compounds in lower case/upper case. Choose one and be consistent in its use.

26. Page 26, line 484: ‘gram-negative’ needs to be replaced with ‘Gram-negative’. ‘Gram’ refers to the surname of the scientist who designed the staining technique and should therefore always be written with a capital letter.

27. Page 28, line 519: Remove the additional full stop after ‘sp.’

28. Page 28, line 527: Remove ‘one’.

29. Even though it was attempted to highlight additional language editing required, it is recommended that the revised manuscript should be evaluated once again before resubmission.

Reviewer #3: The manuscript by Flores and colleagues deals with the isolation and application of actinobacteria recovered from lagoons in Peru. No doubt the MS is interesting but I feel some sections should be improved.

1. It does not seem to be fully details on the culture conditions, time, and morphology of the colonies as these were recovered from the isolation plates and subcultured for further analyses. I think this is important should other authors who read the MS may also be encouraged to perform or follow the protocols indicated on their MS. If this information is missing, then I do not see how other authors may follow or identify their cultures.

2. After sequencing the 16S of the isolates, the authors indicate a size of 1,206 nt but the 16S rDNA gene is nearly 1,500 nt. Is there any particular reason why the gene was only partially sequenced and not completed? I am not saying that the full gene should be sequenced to accept the ms but an explanation within the MS would be useful on why 300nt were missing (technical problems? primers used? sequencing reactions?).

3. Figure 3 indicates: Type strains within the genus Streptomyces and Pseudonocardia but the figure is not related nor include any Pseudonocardia, only Streptomyces!

4. In Figure 2 Streptomyces strain MW562807 falls in a subclade together with many other isolates. However, in that same tree MW562807 is also related to Streptomyces olivaceus and S. pactum. Have the authors checked if those two streptomycetes also show the same or similar biological activities to their isolate? Also since MW562807 is the most interesting strain, why the authors did not include on their study all related isolates to that strain acording to Figure 2? It would definitively be interesting to know if all the isolates within that subclade in Figure 2 do also show similar activities. Again, this is not meant to hold the MS any longer but a few lines within the conclusions or future work could be mention.

5. Conclusion. It is clear that the authors have a good set of interesting strains which have been analysed in the MS. There is the MW562805 and related isolates but, in my view, MS562814 also deserves more comments. That isolate is not related to any type strain and it seems to be a putative novel species as it is not related to any other streptomycete in their phylogenetic trees. If the authors are to continue with this kind of studies, which one they think deserve more attention? The MSW562805 subclade or the MS562814?

7. PLOS authors have the option to publish the peer review history of their article (what does this mean?). If published, this will include your full peer review and any attached files.

Reviewer #1: No

Reviewer #3: No

---

## [Author Response · Author response to Decision Letter 1]

14 Jun 2021

Response to the Editor comments

Comments from editor:

1. Journal Requirements:

Reply: We have corrected the references according to PLOS ONE's requirements.

Response to the reviewer’s comments

Reviewer #1: 

1. Page 2-3, lines 54-55 before, now 47-48 lines: ‘Three Streptomyces sp. strains’ sound cumbersome – consider changing the wording to ‘Three streptomycetes, strains MW562814,…’

 REPLY: Thank you. According to your suggestions, this manuscript we have corrected.

2. Page 3, lines 59-60 before, now 51-52 lines: To streamline the sentence, consider writing ‘… and the multi-drug resistant Acinetobacter baumannii AC-972.’

 REPLY: Thank you. According to your suggestions, this manuscript we have corrected.

3. Page 3, line 61: ‘…in the R2A medium was determined using a…’

REPLY: Thank you. According to your suggestions, the manuscript has been corrected, and the abstract has been reduced for clarity, and this sentence in the manuscript text has been deleted.

4. Page 4, lines 79-81 before, now 67-68 lines: Consider merging the two sentences to provide more flow to the information provided, e.g., ‘…were reported from saline soils of the ephemeral salty lakes in Buryatiya [4], while Streptomyces, Nocardiopsis, and…’

REPLY: Thank you. According to your suggestions, this manuscript we have corrected.

5. Page 4, lines 81-83. The sentence on biological activities seems out of place – I would recommend removing it.

REPLY: Thank you. According to your suggestions, this manuscript we have removing it.

6. Page 4, line 89 before, now 74: ‘…to the production of enzymes, which can…’

REPLY: Thank you. According to your suggestions, this manuscript we have corrected.

7. Page 8, lines 147, 156, 159-160 before, now 137,147, 150: It is conventional to refer to 16S rRNA (as per other sections in the manuscript).

REPLY: Thank you. According to your suggestions, this manuscript we have corrected.

8. Page 8, line 158 before, now 148,149: What is meant by ‘represented’? Did you mean that the sequences were submitted to the EZBioCloud 16S database ‘Identify’ service?

REPLY: Thank you. According to your suggestions, this manuscript we have corrected for “were added”.

9. Page 9, line 162 before, now 152: ‘Mega’ should be written as ‘MEGA’

REPLY: Thank you. According to your suggestions, this manuscript we have corrected.

10. Throughout the manuscript: For any % solutions, please indicate whether they are v/v or w/v solutions.

REPLY: Thank you. According to your suggestions, this manuscript we have corrected.

11. Page 10, line 187 before, now 177,178: What is meant by ‘adjusted to a total volume of 200�l’? And is the ‘plate’ referred to a 96-well plate?

REPLY: Thank you. According to your suggestions, this manuscript we have corrected “total volume of 200�l” and add 96-well plate.

12. Page 13, lines 251-252 before, now 239,240: ‘…were used to group isolates into 42 filamentous and 8 non-filamentous bacterial groups’. Can’t use the information to ‘classify’, rather ‘group’, and since all the isolates were bacteria, best to keep it generic at this stage and use an inclusive term such as ‘bacteria’.

REPLY: Thank you. According to your suggestions, this manuscript we have corrected.

13. Table 2: The Family names should also be written in italics.

REPLY: Thank you. According to your suggestions, this manuscript we have corrected.

14. Page 14, line 283: Here is it indicated that MEGA7 was used, whereas in the methodology section it is indicated that MEGA6 was used – please correct.

REPLY: Thank you. According to your suggestions, this manuscript we have corrected.

15. Page 15, line 298 before, now 269 line: Remove ‘to’

REPLY: Thank you. According to your suggestions, this manuscript we have corrected.

16. Page 15, line 303 before, now 274: Replace ‘Actinobacteria’ with ‘actinobacterium’

REPLY: Thank you. According to your suggestions, this manuscript we have corrected.

17. Page 16, line 310 before, now 280 line: Replace ‘like’ (colloquial term) with ‘such as’ (more scientific).

REPLY: Thank you. According to your suggestions, this manuscript we have corrected.

18. Page 16, line 312 before, now 281 line: Did you mean to say that a member of the genus Pseudonocardia was also recovered during this study? In which case, the wording needs to change to reflect this: ‘…and plants, was also recovered in this study.’

REPLY: Thank you. According to your suggestions, this manuscript we have corrected.

19. Page 16, line 316 before, now 285 line (and throughout the manuscript): If there is no reference made to specific strains of a genus, then it is best to write out spp. as species. For example, here, it is best to write ‘…Streptomyces species as the most…’

REPLY: Thank you. According to your suggestions, this manuscript we have corrected.

20. Table 3: Replace ‘Isolates of Streptomyces sp.’ with ‘Streptomyces isolates’

REPLY: Thank you. According to your suggestions, this manuscript we have corrected.

21. Figure 3: It is recommended that the figure should be removed. The addition of all streptomycete isolates to Figure 2 negated the need for this figure.

REPLY: Thank you. According to your suggestions, this manuscript we have removing it.

22. Page 22: Be consistent in how the names of bioactive compounds are presented, e.g., shellmicin is written in both lower case and upper case in this section (see lines 430 and 436 vs 434 and 436) before, now (384,391,395,396,etc)

REPLY: Thank you. According to your suggestions, this manuscript we have corrected Shellmicin A, …

23. Figure 4: How does the inset information (Streptomyces B-81 and Control) relate to the figure presented?

REPLY: Thank you. According to your suggestions, it was corrected for better understanding, than is indicated in the description in the figure 3.

24. Page 25, line 463 before, now 423,424 lines: What is meant by ‘rolling in the GNPS platform database’?

REPLY: Thank you. According to your suggestions, this manuscript we have added term “comparing in the GNPS platform database”.

25. Page 25: Please see comment above about writing the names of bioactive compounds in lower case/upper case. Choose one and be consistent in its use.

REPLY: Thank you. According to your suggestions, this manuscript we have corrected.

26. Page 26, line 484: ‘gram-negative’ before, now 443 line needs to be replaced with ‘Gram-negative’. ‘Gram’ refers to the surname of the scientist who designed the staining technique and should therefore always be written with a capital letter.

REPLY: Thank you. According to your suggestions, this manuscript we have corrected.

27. Page 28, line 519 before, now 492 line: Remove the additional full stop after ‘sp.’

REPLY: Thank you. According to your suggestions, this manuscript we have corrected.

28. Page 28, line 527 before, now 498 line: Remove ‘one’.

REPLY: Thank you. According to your suggestions, this manuscript we have corrected.

29. Even though it was attempted to highlight additional language editing required, it is recommended that the revised manuscript should be evaluated once again before resubmission.

REPLY: Thank you, if there has been a resignation of the major revision of the language in this new version committed.

Reviewer #3: 

1. It does not seem to be fully details on the culture conditions, time, and morphology of the colonies as these were recovered from the isolation plates and subcultured for further analyses. I think this is important should other authors who read the MS may also be encouraged to perform or follow the protocols indicated on their MS. If this information is missing, then I do not see how other authors may follow or identify their cultures.

REPLY: Thank you for your comment, we appreciate your evaluation very much. This information was described in the protocol on open access data base which was mentioned in the methodology section 132-133 (as described in a previously reported protocol https://www.protocols.io/view/actinobacteria-collection-enrichment-and-isolation-brztm76n). We describe the culture conditions, growth time and type of colonies morphology by microscopic observation.

2. After sequencing the 16S of the isolates, the authors indicate a size of 1,206 nt but the 16S rDNA gene is nearly 1,500 nt. Is there any particular reason why the gene was only partially sequenced and not completed? I am not saying that the full gene should be sequenced to accept the MS but an explanation within the MS would be useful on why 300nt were missing (technical problems? primers used? sequencing reactions?).

 REPLY: Thank you. The primers used amplify about 1000 bp covering regions V1 to V8, whose variability for genera level distinction is commonly used in diversity studies. It was added in the description of the phylogenetic tree in the specification if sequenced with primer 10F to 1492R, which resulted in cured sequences between 1206-1500 bp.

3. Figure 3 indicates: Type strains within the genus Streptomyces and Pseudonocardia but the figure is not related nor include any Pseudonocardia, only Streptomyces!

 REPLY: Thank you. This figure has been replaced to include the genera Streptomyces and Pseudonocardia.

4. In Figure 2 Streptomyces strain MW562807 falls in a subclade together with many other isolates. However, in that same tree MW562807 is also related to Streptomyces olivaceus and S. pactum. Have the authors checked if those two streptomycetes also show the same or similar biological activities to their isolate? Also since MW562807 is the most interesting strain, why the authors did not include on their study all related isolates to that strain acording to Figure 2? It would definitively be interesting to know if all the isolates within that subclade in Figure 2 do also show similar activities. Again, this is not meant to hold the MS any longer but a few lines within the conclusions or future work could be mention.

 REPLY: Thank you. According to your suggestions, this manuscript, were added three new references of Lobophorins compounds isolated from Streptomyces olivaceus strains (on lines 461-475), and it is presented that the strains that are reported for their biological activity from different lobophorins are also close S. olivaceus and S. pactum. In this study, a representative clade isolate was chosen to assess biological activities, however, we do not rule out this potential in future studies.

5. Conclusion. It is clear that the authors have a good set of interesting strains which have been analysed in the MS. There is the MW562805 and related isolates but, in my view, MS562814 also deserves more comments. That isolate is not related to any type strain and it seems to be a putative novel species as it is not related to any other streptomycete in their phylogenetic trees. If the authors are to continue with this kind of studies, which one they think deserve more attention? The MSW562805 subclade or the MS562814?

 REPLY: Thank you. According to your suggestions, this manuscript we could not only discriminate a single strain, because all three have bioactive activities that is why the three will be taken to future work as we have mentioned in the conclusion within the lines (508-512). Taxonomic studies to characterize the strain MS562814 have been conducted by the group to describe a possible new species of the genus Streptomyces.

---

## [Editor Report · Decision Letter 2]

16 Jul 2021

EVALUATION OF ANTIMICROBIAL AND ANTIPROLIFERATIVE ACTIVITIES OF ACTINOBACTERIA ISOLATED FROM THE SALINE LAGOONS OF NORTHWEST PERU

PONE-D-20-29463R2

Dear Dr. Flores Clavo,

We’re pleased to inform you that your manuscript has been judged scientifically suitable for publication and will be formally accepted for publication once it meets all outstanding technical requirements.

Despite the "acceptance" of your Manuscript, there are a few extra comments that should be addressed together with the technical requirements. A list of such minor comments which are merely mine follows:

1. The abstract should be shortened 3-5 lines. The chemical details can be easily avoided in the abstract.

2. Line 158, it says "on closest hits" and should be "on the closest hits"

3. Lines 277, 278, 281, 282, 284 285 and 286. All of these lines use the word "*Streptomyces*" but this paragraph is mainly devoted to *Streptomyces* hence it is redundant to re-write *Streptomyces* for each species and the corresponding comparisons. I suggest you simply use "S." which it is understood, due to the context of the paragraph, referring to *Streptomyces*.

4. Line 292. It currently says "has recovered" but I think it should be "was recovered".

5. Line 354. It says "*Kitasospora*" but it should be "*Kitasatospora*". Despite this typographical mistake I'm not quite sure if this outgroup would be the best because *Kitasatospora* belongs to the same family as *Streptomyces*. Perhaps a total different actinobacteria such as *Micromonospora* which is not related neither to *Pseudonocardia* nor *Streptomyces* would be better.

I strongly believe that these minor corrections should contribute to the improvement of the manuscript hence I kindly ask you to follow them together with the technical requirements of the editing process.  

Kind regards,

Luis Angel Maldonado Manjarrez, Ph.D.

Academic Editor

PLOS ONE

---

## [Editor Report · Acceptance letter]

27 Aug 2021

PONE-D-20-29463R2 

Evaluation of antimicrobial and antiproliferative activities of actinobacteria isolated from the saline lagoons of northwestern Peru 

Dear Dr. Flores Clavo:

I'm pleased to inform you that your manuscript has been deemed suitable for publication in PLOS ONE. Congratulations! Your manuscript is now with our production department. 

Kind regards, 

on behalf of

Dr. Luis Angel Maldonado Manjarrez 

Academic Editor

PLOS ONE